# B cell zone reticular cell microenvironments shape CXCL13 gradient formation

Jason Cosgrove[1,2,3,19], Mario Novkovic [4,19], Stefan Albrecht[5,19], Natalia B. Pikor[4], Zhaoukun Zhou[6,7,8], Lucas Onder[4], Urs Mörbe[4], Jovana Cupovic[4], Helen Miller [6,7,8], Kieran Alden[1,3], Anne Thuery[2], Peter O'Toole[6], Rita Pinter[9], Simon Jarrett[9], Emily Taylor[2], Daniel Venetz[10], Manfred Heller[11], Mariagrazia Uguccioni[10], Daniel F. Legler [12], Charles J. Lacey [1], Andrew Coatesworth[13], Wojciech G. Polak[14], Tom Cupedo[15], Bénedicte Manoury[16,17], Marcus Thelen[10], Jens V. Stein [18], Marlene Wolf [5], Mark C. Leake [6,7,8✉], Jon Timmis[1,3✉], Burkhard Ludewig [4✉] & Mark C. Coles[1,9✉]

Through the formation of concentration gradients, morphogens drive graded responses to extracellular signals, thereby fine-tuning cell behaviors in complex tissues. Here we show that the chemokine CXCL13 forms both soluble and immobilized gradients. Specifically, CXCL13+ follicular reticular cells form a small-world network of guidance structures, with computer simulations and optimization analysis predicting that immobilized gradients created by this network promote B cell trafficking. Consistent with this prediction, imaging analysis show that CXCL13 binds to extracellular matrix components in situ, constraining its diffusion. CXCL13 solubilization requires the protease cathepsin B that cleaves CXCL13 into a stable product. Mice lacking cathepsin B display aberrant follicular architecture, a phenotype associated with effective B cell homing to but not within lymph nodes. Our data thus suggest that reticular cells of the B cell zone generate microenvironments that shape both immobilized and soluble CXCL13 gradients.

[1] York Computational Immunology Lab, University of York, York, UK. [2] Centre for Immunology and Infection, Department of Biology and Hull York Medical School, University of York, York, UK. [3] Department of Electronic Engineering, University of York, York, UK. [4] Institute of Immunobiology, Kantonsspital St. Gallen, St. Gallen, Switzerland. [5] Theodor Kocher Institute, University of Bern, Bern, Switzerland. [6] Department of Biology, University of York, York, UK. [7] Biological Physical Sciences Institute (BPSI), University of York, York, UK. [8] Department of Physics, University of York, York, UK. [9] Kennedy Institute of Rheumatology at the University of Oxford, Oxford, UK. [10] Institute for Research in Biomedicine, Università della Svizzera italiana, Bellinzona, Switzerland. [11] Department of Clinical Research, University of Bern, Bern, Switzerland. [12] Biotechnology Institute Thurgau (BITg) at the University of Konstanz, Kreuzlingen, Switzerland. [13] York Teaching Hospital NHS Foundation Trust, York, UK. [14] Department of Surgery, Erasmus University Medical Centre, Rotterdam, Netherlands. [15] Department of Hematology, Erasmus University Medical Centre, Rotterdam, Netherlands. [16] Institut Necker Enfants Malades, INSERM U1151- CNRS UMR 8253, 149 rue de Sèvres 75015 Paris, France Université René Descartes, 75005 Paris, France. [17] Université Paris Descartes, Sorbonne Paris Cité, Paris, France. [18] Department of Oncology, Microbiology and Immunology, University of Fribourg, Fribourg, Switzerland. [19] These authors contributed equally: Jason Cosgrove, Mario Novkovic, Stefan Albrecht. ✉email: mark.leake@york.ac.uk; jon.timmis@sunderland.ac.uk; burkhard.ludewig@kssg.ch; mark.coles@kennedy.ox.ac.uk

Nonhematopoietic stromal cells regulate the development and maintenance of niches within lymphoid tissues to support the retention, activation, and proliferation of adaptive immune cells in response to antigenic stimulation[1–4]. In the context of antibody-mediated immunity, B cells must migrate to the follicle where they (i) acquire and process antigen; (ii) present antigen to CD4+ T helper cells; and (iii) organize into a germinal center (GC)[5]. Through the secretion of signaling molecules, fibroblastic reticular cells orchestrate both trafficking of B cells to and within different tissue subcompartments, with dysregulation of migration leading to defective follicular homing[6,7], aberrant follicular and GC organization[7,8], and GC-derived lymphomas[9].

Despite the importance of these migratory cues, the distances and scales over which they act are unclear. Many studies suggest that soluble factors, such as the cytokine IL-2, are spatially regulated through a diffusion−consumption mechanism that creates a concentration gradient capable of fine-tuning cell behaviors through a graded exposure to ligand[10]. Consistent with the source-sink scheme of gradient formation, atypical chemokine receptor 4-expressing lymphatic endothelial cells (LECs) lining the ceiling of the subscapular sinus have been implicated in the formation of functional CCL21 chemokine gradients in the lymph node[11]. Interestingly, both molecules are known to dynamically interact with extracellular matrix (ECM) components such as glycosaminoglycans (GAGs)[12–15]. Many soluble factors have carbohydrate-binding domains, a feature that may limit the capacity to undergo free diffusion, particularly in dense tissues[12–14,16,17].

For many molecules, the ability to bind ECM components is a key determinant of functionality[18,19]. In vivo, truncation of the highly charged C-terminus of CCL21 prevents its immobilization to high endothelial venules (HEVs) while the mutant forms of CC chemokines that lack GAG-binding domains fail to induce chemotaxis into the peritoneum[18,20]. Mice carrying a mutated form of CXCL12 (CXCL12$^{gagtm}$) where interactions with the ECM are impaired have disorganized GCs, as well as having fewer somatic mutations in immunoglobulin genes[21]. These experimental studies are supported by mathematical analyses predicting that gradient formation is increased when chemokines are secreted in matrix-binding form as compared to a non-matrix-interacting form[22]. This dichotomy has been explicitly studied in the context of CCL21, where immobilized and soluble gradients promote adhesive random migration or chemotactic steering of dendritic cells, respectively[15].

In this study we focus on the chemokine CXCL13, a small globular protein with a theoretical average mass of 10.31 kDa that has emerged as a key regulator of B-cell migration and lymphoid tissue architecture, with CXCL13$^{-/-}$ mice displaying aberrant follicular organization[7,23,24]. Similarly, mice deficient in CXCR5, the cognate receptor for CXCL13, have defective formation of primary follicles and GCs in the spleen, with B cells failing to home effectively to the follicles[6,7]. CXCL13 bioavailability is a dynamic function of production, diffusion, immobilization, mobilization, and consumption[25]. Consequently, the precise localization of CXCL13 within lymphoid tissues is difficult to visualize directly.

During selective ablation of follicular reticular cells, also known as follicular dendritic cells (FDCs), follicles remodel into disorganized bands of B cells that retain CXCL13-expressing stromal cell populations[3], suggesting that the cellular sources of this molecule are heterogeneous[4]. The expression patterns of CXCL13 also vary temporally over the course of immunization and infection. Expression is regulated in a positive feedback loop involving CXCR5-mediated induction of LTα$_1$β$_2$ expression by B cells which in turn contributes to maximal CXCL13 production[7].

Once secreted, CXCL13 must diffuse through a dense environment comprising lymphocytes, reticular cells, vasculature, lymphatics, and ECM before undergoing internalization by typical and atypical chemokine receptors or protease-mediated enzymatic degradation[11,26,27]. CXCL13 has been shown experimentally to interact with heparan sulfate via two distinct binding interfaces[17]. Consistent with this structural study, recent single-molecule imaging measurements of chemokine diffusion in ex vivo murine tissue sections and collagen matrices suggest that chemokines may be heterogeneous in their mobility behaviors, with CXCL13 diffusion tightly constrained in tissues[28]. An additional layer of complexity is added by the heterogeneous distribution of ECM proteins within the follicle[29] and by altered chemotactic potency of many chemokines following proteolytic cleavage[30,31]. A number of proteases are known to alter chemokine activity including matrix metalloproteinases, dipeptidylpeptidase IV (CD26), aminopeptidase N (CD13), neutrophil granule proteases, and members of the cathepsin family[30,31]. However, the role of proteolytic processing in the context of gradient formation in vivo is poorly understood.

Given the complexity of the CXCL13 regulatory network, it is unclear if the molecule acts in an immobilized or soluble form and whether proteolytic processing is required to modulate CXCL13 function in vivo. This limited understanding is exacerbated by a dearth of experimental techniques capable of manipulating molecular gradients in situ. Our aim is to understand the mechanisms that create CXCL13 gradients within the B-cell follicle. Here, we employ a modeling and simulation approach, mapping the reticular cell architecture of the primary follicle and reconstructing it in silico. We then apply a combination of machine learning and optimization approaches to systematically generate different chemotactic gradients and assess associated B-cell scanning rates. Using this approach, it is possible to obtain insights where direct experimentation is intractable, generating data with high spatial and temporal sensitivity across multiple scales of organization.

Using a modeling and simulation approach, in combination with imaging and biochemistry, we assess the mechanisms that regulate CXCL13 gradient formation within lymphoid tissues. Our integrative approach shows that within the follicle, CXCL13 can exist in a soluble or immobile form. CXCL13 solubilization is regulated by the protease cathepsin B (Ctsb), with cleaved CXCL13 showing altered binding kinetics and increased chemotactic potency. Strikingly, in Ctsb-deficient mice, B-cell localization is highly variable, with an increased propensity to form ring-like structures around the T-cell zone, suggesting a key role for soluble CXCL13 in follicle formation. Our data thus suggest that reticular cells of the B-cell zone generate microenvironments that shape both immobilized and soluble CXCL13 gradients.

## Results

**Mapping CXCL13$^{±}$ stromal cell networks in the B-cell follicle.**
In this study we couple experimental and modeling approaches to identify and enumerate key entities and processes that regulate CXCL13 bioavailability (Supplementary Fig. 1). To understand the cellular sources of CXCL13 within the primary follicle, we mapped the 3-dimensional (3D) organization of CXCL13+ stromal cells in lymph node tissue sections from Cxcl13-Cre/Tdtomato R26R-EYFP (abbreviated as Cxcl13-EYFP) mice[4]. In Cxcl13-EYFP mice, EYFP acts as a lineage marker, endogenously expressed in cells that originate from a CXCL13-producing precursor, while TdTomato expression (red fluorescent protein, RFP) is confined to cells with current CXCL13 promoter activity. In addition, we identify FDCs as cells that are also CD21/35 positive (Fig. 1a). From a follicle tissue cross-section, we mapped

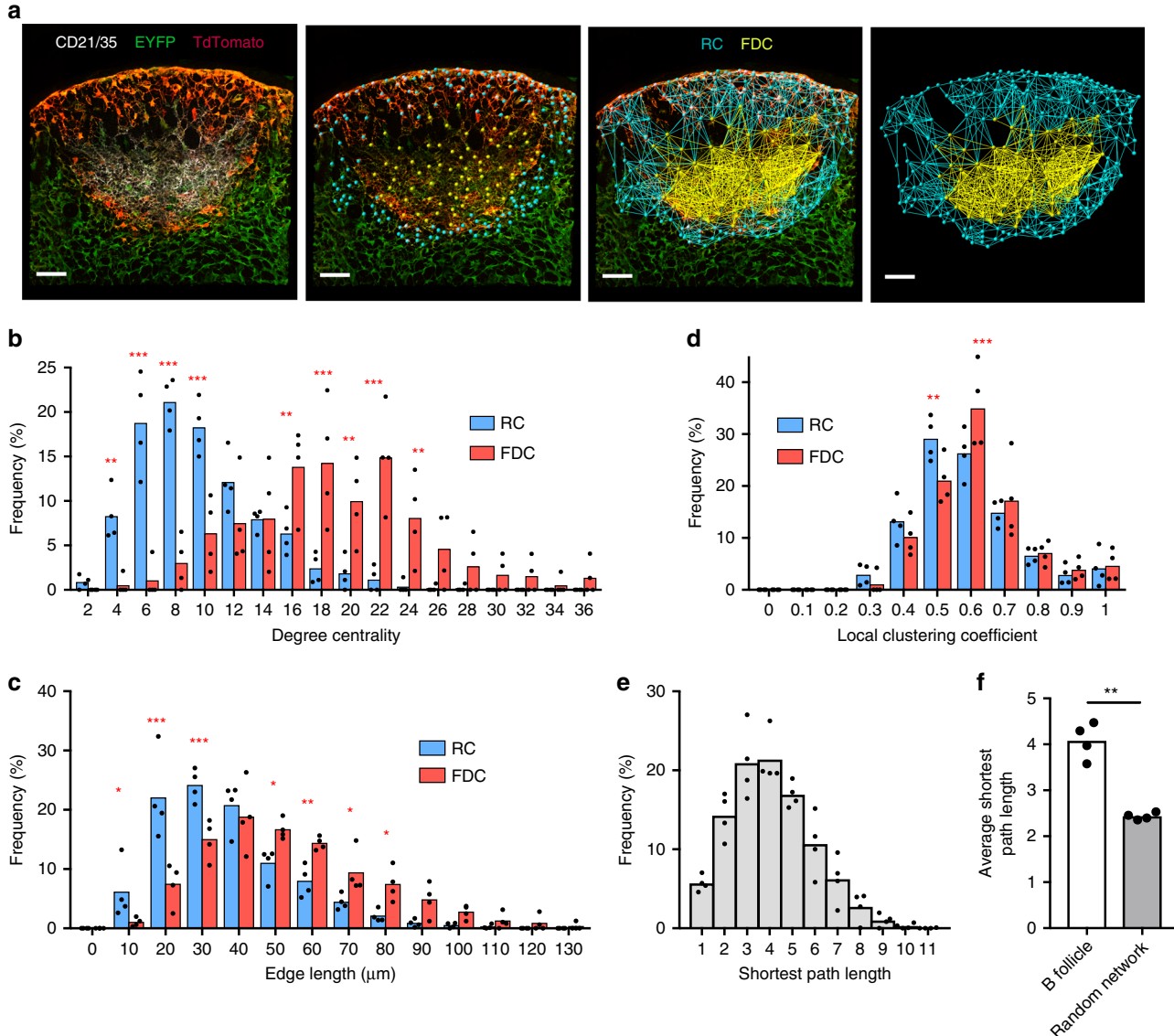

**Fig. 1 The topological network properties of CXCL13$^+$ follicular stromal cells. a** Mapping confocal images of lymph node follicles taken from Cxcl13-cre/EYFP reporter mice using the Imaris image analysis software. The FDC subnetwork is highlighted in yellow and the RC subnetwork in cyan. Distributions of degree centrality, edge length and local clustering coefficient are indicated for the FDC and RC subnetworks (**b−d**). **e** Distribution of shortest path lengths is indicated for the global follicular network and are compared to that of an equivalent random network with the same number of nodes and edges (**f**). Data represent mean ± SD for $n = 4$ mice. Statistical significance was determined using a two-way ANOVA with Sidak's multiple comparison test. *$p < 0.05$, **$p < 0.01$, ***$p < 0.001$. Scale bar = 50 µm. Source data are provided as a Source Data file.

a network of 198 ± 39 nodes and 1163 ± 242 edges ($n = 4$ mice), whereby we define nodes as the EYFP$^+$RFP$^+$ reticular cells (RCs) and FDCs, while edges are indicated as physical connections between neighboring nodes (Fig. 1a, Supplementary Table 1). We subdivide CXCL13$^+$ follicular reticular cells into two broad categories: CXCL13$^+$ CD21/35$^+$ FDCs and CXCL13$^+$ CD21/35$^-$ reticular cells (CD21$^-$ RCs) comprising reticular cells located underneath the subcapsular sinus (marginal reticular cells), and at the outer follicle. Interestingly, FDCs display significantly higher degree centralities and edge lengths than CD21$^-$ RCs, forming a dense subnetwork within the follicle (Fig. 1b, c). Topological analysis (as described in Supplementary Note 1) of the clustering coefficients ($C_{global} = 0.57 ± 0.02$, $C_{local} = 0.60 ± 0.02$) and the average shortest path length ($4.17 ± 0.26$) through the network has revealed a significant difference in the topological organization of the follicle network as opposed to an equivalent random network with the same number of nodes and edges ($C_{local} = 0.06 ± 0.01$, $C_{global} = 0.06 ± 0.01$ and shortest path length = $2.41 ± 0.11$). These results indicate that the follicle network exhibits small-world properties (Fig. 1d, e) reminiscent of the T-cell zone FRC network[32]. These findings are further corroborated by comparing the follicle network to an idealized small-world network (WS), demonstrating their similarity in topological organization and small-world network metrics $\sigma$ and $\omega$ (Supplementary Table 1). The small-world configuration is characterized by an overabundance of highly connected nodes, common connections mediating the short mean-path lengths. This property is associated with rapid information transfer and is also observed in airline routes and social networks[33,34]. In the context of the follicle, this property is likely to promote complement-mediated trafficking of antigen by non-cognate B cells from the subcapsular sinus to the FDC network, and also the migration of cognate B

cells as they search for antigen within the follicle, and then present it to T cells at the interfollicular border before seeding a GC reaction[5,35,36].

**Simulating and optimizing CXCL13 gradients in silico**. Since the structural organization of CXCL13$^+$ reticular networks are a key determinant of follicle functionality, we hypothesized that that this cellular architecture may also regulate the molecular level patterning of CXCL13. To address this hypothesis, we use the stromal cell topology dataset to inform an algorithmic reconstruction of the follicular reticular cell network in silico[37]. Coupled with additional imaging datasets (Supplementary Fig. 1), we engineered a high fidelity (Supplementary Fig. 3) multiscale representation of the primary follicle in which immune cell agents can interact with reticular cells, creating and shaping complex physiological CXCL13 gradients (Fig. 2a, Supplementary Note 3). This quantitative approach facilitates simulation analysis of CXCL13 gradient formation at very high spatiotemporal resolution but does require significant computational resources to evaluate (detailed in Supplementary Note 2), limiting the range of analysis techniques we can apply to understand CXCL13 gradient formation. To address this issue, we complemented our simulation analysis with an emulation-based approach (Fig. 2b, Supplementary Fig. 2). In this approach a machine-learning algorithm known as an artificial neural network (ANN) was used to learn the emergent behaviors of the simulator, such that it was capable of rapidly and accurately mapping between simulation inputs and outputs averaged over a high number of replicate runs (Fig. 2b, Supplementary Fig. 2).

To assess whether CXCL13 acts in principally an immobilized or a soluble form, we focused on two potential models: Model 1 suggests that CXCL13 binds to ECM components creating short sharp gradients proximal to the CXCL13-secreting cells, while in model 2 where CXCL13 is largely soluble and diffuses more freely throughout the tissue, creating a more homogeneous pattern (Fig. 2c). To assess the veracity of each theory, a chemotactic landscape was created for each model through tuning parameters that control the rate of secretion, diffusion, and decay but keep overall concentration fixed and the emergent scanning rates of in silico B cells were quantified under each scenario. This analysis predicted that model 1 yields higher scanning rates than model 2, suggesting that model 1 is more likely (Fig. 2d). To further assess the veracity of this result, we perform an optimization analysis to determine the most effective spatial distribution of CXCL13 with respect to antigen scanning. In this analysis we employed the nondominated sorting genetic algorithm-II (NSGA-II)[38,39] to systematically perturb parameters relating to CXCL13 bioavailability in silico and determine a Pareto front of solutions (emergent cell migration behaviors) that represent the best trade-off obtained between fitting experimentally determined migration patterns (objectives 1−3, detailed in "Methods")[40] and maximizing scanning rates (objective 4, detailed in "Methods")[40]. Despite using a heuristic approach, performing this analysis on our multiscale simulator is computationally intensive due to: (i) a highly complex search space; (ii) the need for replicate runs to mitigate stochastic uncertainty; and (iii) multiple, conflicting objectives. To address this, we combined NSGA-II with our ANN-based emulator, an approach to determine the precise spatial distribution of CXCL13 that would not only fit our data, but also lead to optimal B-cell scanning rates. This approach allowed us to examine the distributions of parameter values that give rise to our optimal solutions, such that we can mechanistically understand why some spatial patterns are more effective than others. More specifically, we found that values of the CXCL13 diffusion constant are skewed towards low values, and

decay rates skewed towards high values (Fig. 2e), consistent with model 1. In addition, we find that our objectives are conflicting, with increased scanning rates leading to poorer agreement between emergent cell behaviors in silico and laboratory measures (Fig. 2f). Our theoretical analysis predicts that immobilized CXCL13 gradients are a key determinant of B-cell trafficking patterns within the follicle.

**CXCL13 forms immobilized gradients within the B-follicle**. To assess our theoretical prediction that CXCL13 can form immobilized gradients, we quantify binding of CXCL13$^{AF647}$ to tonsil tissue sections incubated with heparinase-II, an enzyme that cleaves both heparin and heparan sulfate or phosphate-buffered saline (PBS) (Fig. 3a). By quantifying the fluorescent intensity for each image, we determine a significant drop in fluorescence intensity following heparinase-II treatment, suggesting that CXCL13 binds heparin and/or heparan sulfate in lymphoid tissue follicles (Fig. 3b). To assess whether heparin and heparan sulfate constrain diffusion, we image CXCL13$^{AF647}$ diffusion within CD19$^+$ B-cell follicles of tonsil tissue sections and quantify mobility with super-resolution precision at ∼500 Hz[41] (Fig. 3c). Consistent with simulation analysis and immunohistochemistry, we find that CXCL13$^{AF647}$ is largely immobile yielding a median [IQR] diffusion rate of 0.19 [0.001−0.79] μm$^2$ s$^{−1}$, while treatment with heparinase-II led to increased rates of diffusion with a sample median [IQR] diffusion coefficient of 1.6 [0.47−3.9] μm$^2$ s$^{−1}$ ($p < 0.0001$) (Fig. 3c).

Our super-resolution imaging assay permitted the tracking of single CXCL13 molecules, allowing us to characterize the heterogeneity of CXCL13 mobility in situ. Specifically, we identified and characterized distinct matrix bound (low-mobility) and soluble (high-mobility) fractions (Fig. 3d). Relative to the immobile fraction only a very small proportion of CXCL13 was soluble, consistent with theoretical results. Disruption of the ECM through heparinase-II treatment did lead to an increase in the mobile fraction. Given that such a large proportion of CXCL13 was immobilized, we assessed whether we could detect an immobile CXCL13 fraction within B-cell follicles using immunohistochemistry in fixed human tonsil and lymph node sections (Fig. 4a). The spatial distribution of CXCL13 immunoreactivity is spatially heterogeneous and strongly colocalized with the FDC marker CD35 (Fig. 4a). To quantify this observation, we measure the spatial autocorrelation of CXCL13 expression in tonsil sections and determine the distance ($D_{uncorrelated}$) at which there is no statistically significant correlation in fluorescence intensities (Fig. 4b, c). This analysis indicates that CXCL13 expression is significantly correlated over short distances (∼50 μm) before becoming significantly uncorrelated with no statistically significant difference in $D_{uncorrelated}$ between human tonsils and model 1, corroborating our theoretical observation that CXCL13 can form complex immobilized gradients in the follicle (Fig. 4b, c). These data show that CXCL13 interacts readily with ECM components, and together with stromal-cell network architecture, shapes complex immobilized CXCL13 gradients within the B-cell follicle.

**Cathepsin B generates soluble CXCL13 gradients**. Given the high affinity with which CXCL13 binds to the ECM, we hypothesized that it may undergo proteolytic processing. In this study we focus on the cathepsin family; most cathepsins identified in humans are lysosomal enzymes involved in metabolic protein turnover but many cathepsins have also been reported to cleave chemokines[30,31]. In particular, we have focused our attention on cathepsin B (Cath-B), which has been shown to regulate cytokine expression during *Leishmania major* infection[42], is upregulated in

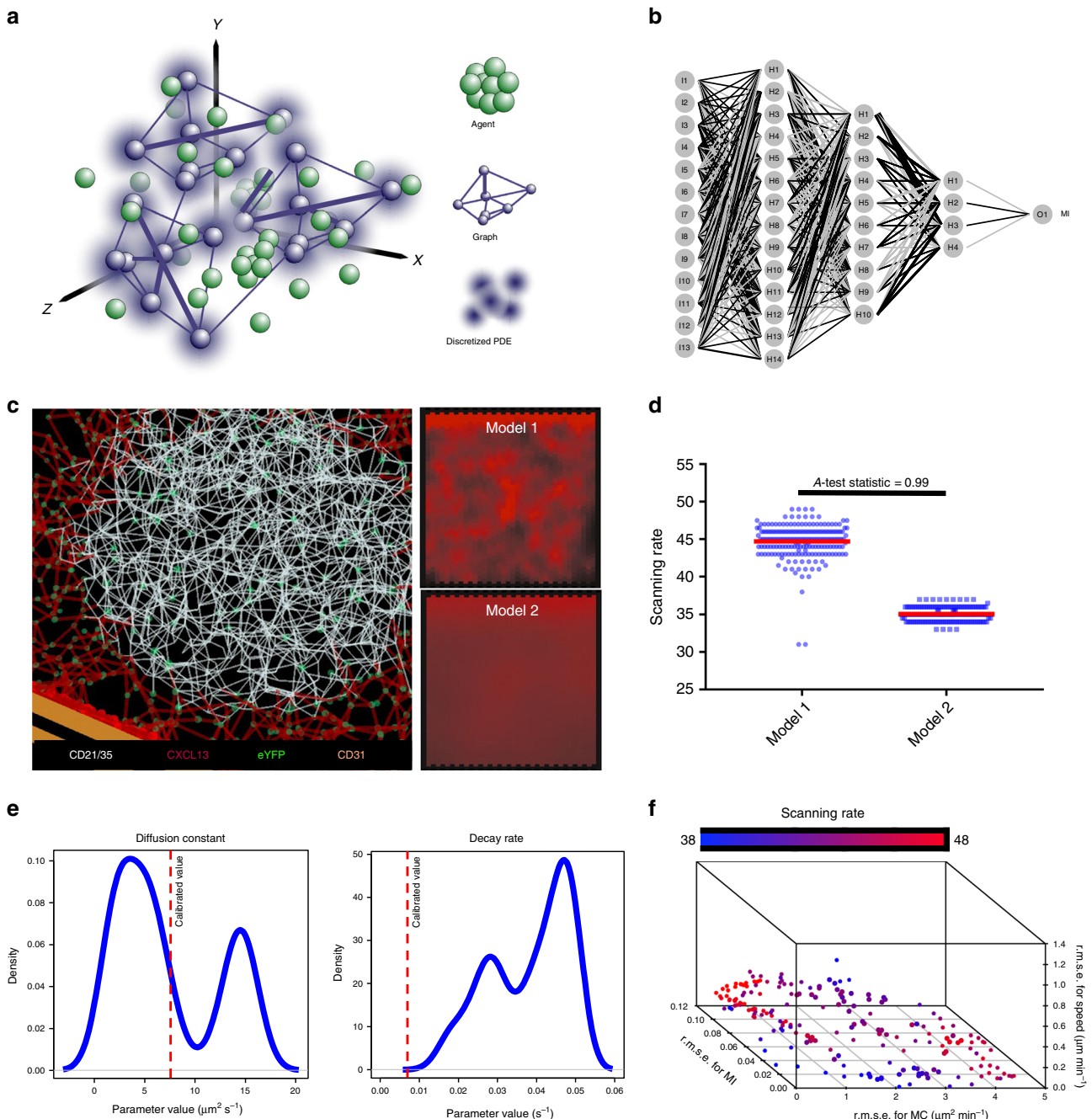

**Fig. 2 Mapping CXCL13 spatial distribution through simulation analysis and multiobjective optimization. a** Overview of the multiscale model platform. In this modular system stromal cells are modeled as a graph (Module 1), chemokine diffusion is modeled as a discretized partial differential equation (Module 2), while B cells are modeled as agents that can interact with their local environment through a set of coupled differential equations and vector-based calculations (Module 3). **b** Example structure of an artificial neural network used to emulate CXCL13Sim. The network has 13 input nodes that connect to three hidden layers, and a single output node predicting the meandering index. A distinct network is created for each simulator output. The hyperparameters of the network were determined using *k*-folds cross-validation. **c** The in silico follicular stromal network with a chemotactic landscape created for models 1 and 2 by the network. **d** Comparison of scanning rates in silico for models 1 and 2. Each parameter set was run 200 times with significance assessed using the Vargha−Delaney A-test[70]. The test statistic (0.99) exceeds the threshold for a large effect size (0.71). Bar plots represent the median value for the emergent scanning rate and the error bars represent the IQR. **e** Parameter distributions for diffusion and decay rates corresponding to the Pareto optimal solutions shown in (**f**) with calibrated values for each parameter shown using the dotted red line. **f** Using a MOEA scheme we seek to address the following four objectives: minimize the root mean squared error between emulator and simulator responses for cell speed, meandering index and motility coefficient; and maximize scanning rates. The Pareto front of solutions represents the trade-off in performance between cell behaviors and scanning rates, using NSGA-II (emulation pipeline described in Supplementary Fig. 1). Source data are provided as a Source Data file.

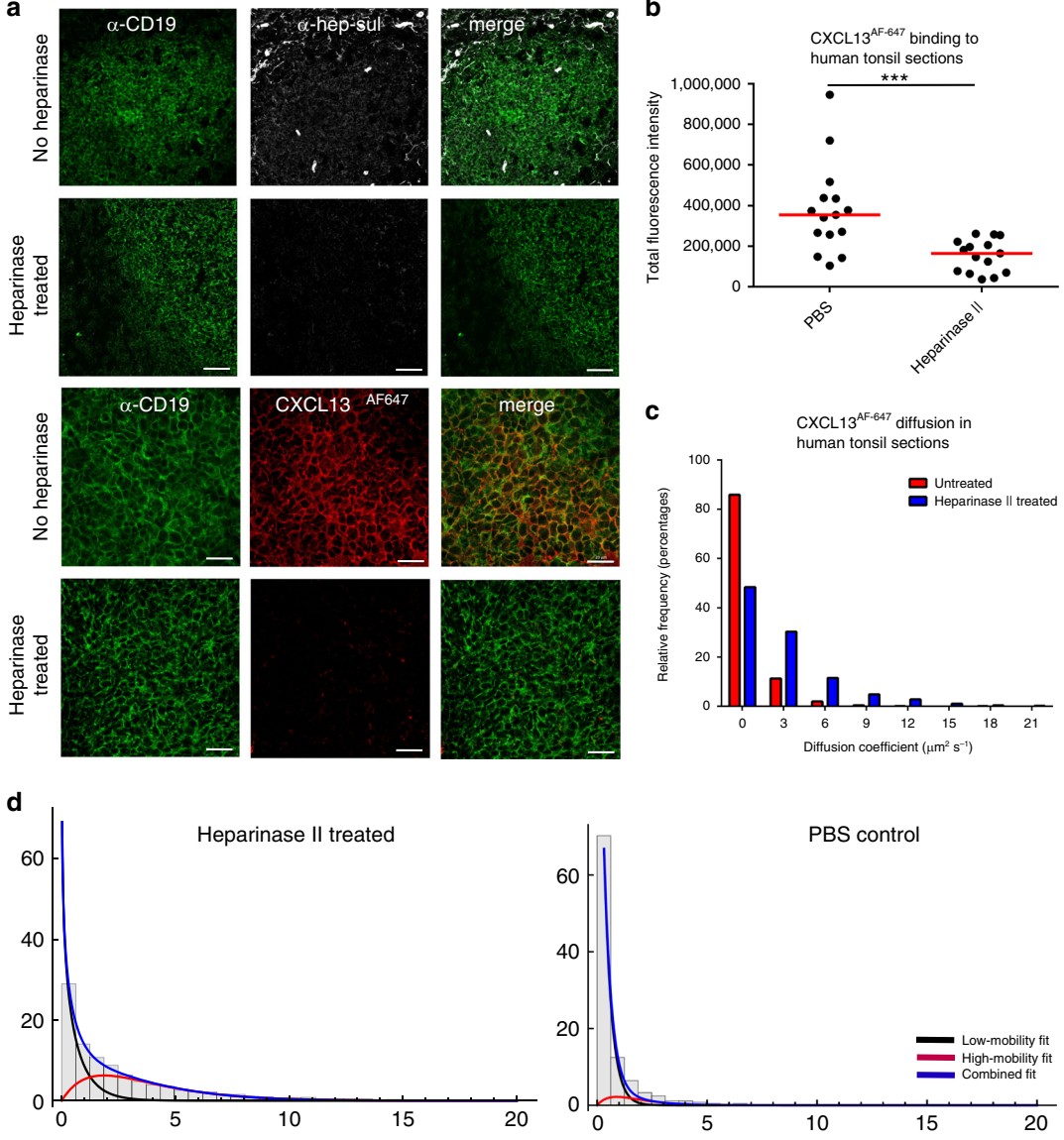

**Fig. 3 CXCL13 interactions with ECM components constrain mobility. a** Tonsil tissue sections were stained with anti-CD19 and anti-heparan sulfate antibodies. Following incubation in PBS or heparinase II treatment binding of CXCL13$^{AF647}$ to the B follicle was assessed. **b** Quantification of total fluorescent intensity for each image. Shapiro−Wilk tests indicated that the datasets were not normally distributed ($p$ value < 0.001) and so significance was assessed using a Mann−Whitney $U$ test ($p$ value < 0.001; ***). Data shown are from a single experiment (from a total of two independent experiments) with each data point representing a distinct follicle obtained from a single patient. **c** Quantification of CXCL13$^{AF647}$ mobility in CD19$^+$-positive regions of human tonsil sections. Diffusion measured in untreated tissue sections is indicated in red with values obtained for heparinase II-treated sections indicated in blue. All tissue sections were obtained from the same patient. The median [IQR] diffusion rate of CXCL13$^{AF647}$ in untreated sections was calculated as 0.19 [0.001−0.79] μm$^2$ s$^{-1}$, while treatment with heparinase-II led to a significantly different (assessed using the Mann−Whitney $U$ test) diffusion coefficient of 1.6 [0.47−3.9] μm$^2$ s$^{-1}$ ($p$ < 0.0001). **d** Characterizing the multiple modes of diffusion observed in our single-molecule tracking analysis in B-follicles treated with heparinase II, or PBS. Source data are provided as a Source Data file.

many cancers[43], and can be produced in extracellular form in cytokine-stimulated fibroblasts taken from rheumatoid arthritis patients[44].

Incubation of CXCL13 with Cath-B yielded two cleavage products with masses of 9.03 and 8.68 kDa, respectively (Fig. 5a). The smaller product is stable and forms across a range of enzyme substrate ratios in both humans and mice (Supplementary Fig. 4a) and is detected at pH values between 4.0 and 7.2 with an optimal turnover rate between pH 5.0 and 6.5 (Supplementary Fig. 4b). Consistent with these data, single-molecule imaging of CXCL13 [1–72] diffusion in 15% Ficoll showed a higher mobility rate for the Cath-B-treated form of the molecule as compared to

untreated (1.0 [0.04−3.6] μm$^2$ s$^{-1}$ and 0.61 [0.08−2.2] μm$^2$ s$^{-1}$ respectively, $p$ < 0.001), indicating that the fluorescent tag incorporated into the C-terminus of the molecule had been cleaved (Supplementary Fig. 4c).

To compare the heparin-binding capacity of CXCL13 and CXCL13[1–72], we loaded both peptides on a HiTrap heparin column followed by elution with an increasing concentration of NaCl. CXCL13[1–72] displays lower heparin-binding affinity and eluted at 0.53 M NaCl (Fig. 5b, peak 2) compared to intact CXCL13, which elutes at 0.62 M NaCl (Fig. 5b, peak 3). To assess if GAG-binding would protect CXCL13 from being proteolysed by Cath-B, we performed cleavage assays in the presence of

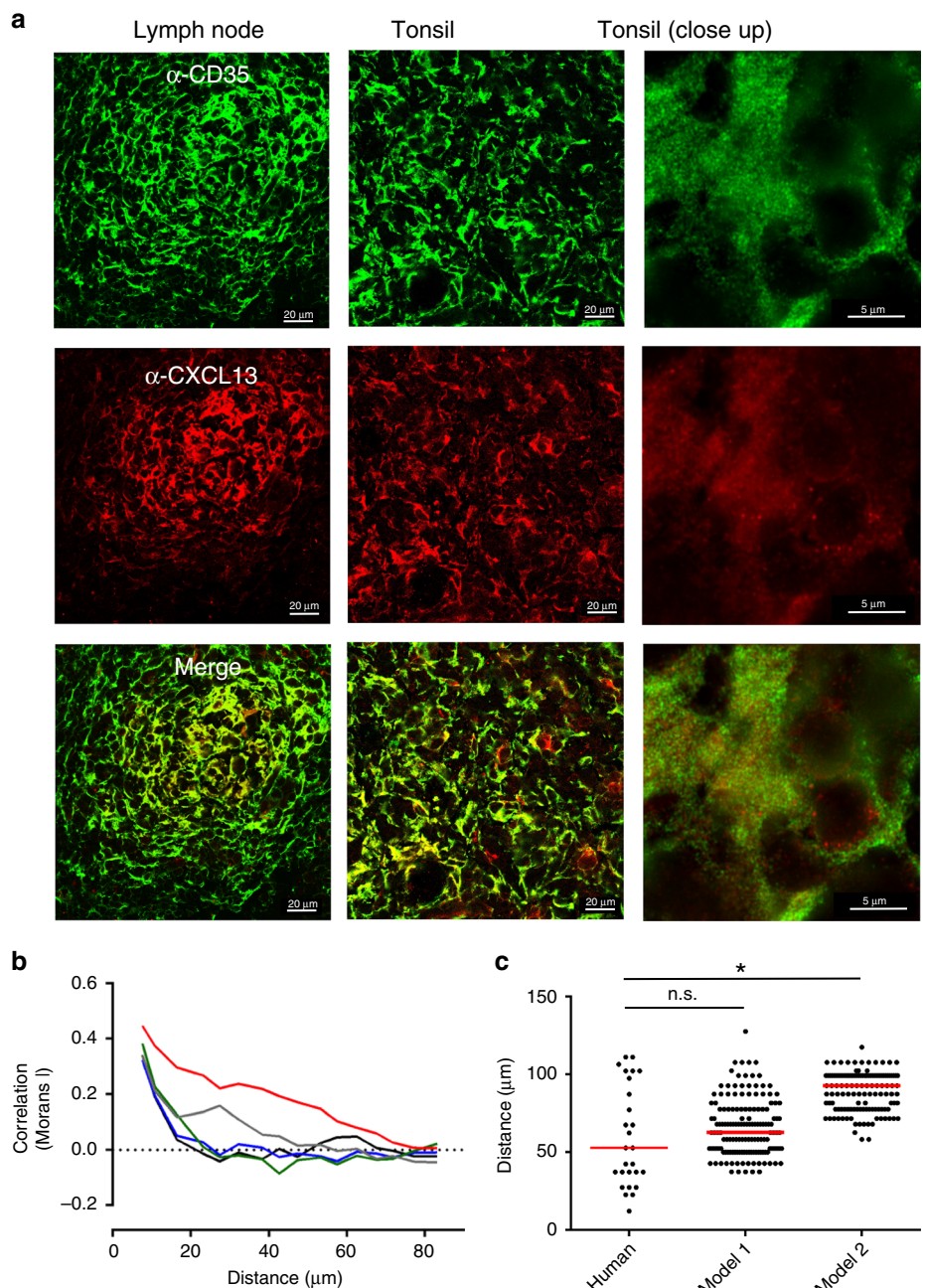

**Fig. 4 Analyzing the spatial distribution of the immobile CXCL13 fraction. a** IHC staining of the FDC marker CD35 (green) and CXCL13 (red) in human lymph nodes and tonsils. **b** The spatial autocorrelation of CXCL13 expression in samples from one patient, each line represents the spatial autocorrelation for a distinct follicle. **c** Comparison of the distances at which no statistically significant spatial autocorrelation (determined using permutation testing as described in "Methods") was detected in human tonsils, and for models 1 and 2. Each data point represents the distance at which no statistically significant spatial autocorrelation was observed for the intensity of anti-CXCL13 staining in a distinct tonsil follicle, with data pooled from five different patients. The red line represents the median distance for each group with significance, the human dataset and each simulation model (run with 200 repeat executions) assessed using the Mann−Whitney $U$ test ($p$ value = 0.06 for model 1 and $p < 0.001$ (denoted as *) for model 2). Source data are provided as a Source Data file.

different GAGs including hyaluronic acid, heparan sulfate, and chondroitin sulfate. The presence of a 5- or 10-fold (w/w) excess of these GAGs, however, does not prevent CXCL13 processing by Cath-B (Fig. 5c). In addition, we stained tonsil sections with an antibody against CXCL13 and quantify the total fluorescent intensity of each image following treatment with Cath-B or PBS (Supplementary Fig. 5). Compared to PBS treatment, incubation with Cath-B led to a statistically significant reduction in the intensity of CXCL13 signal. In conclusion, GAGs do not affect Cath-B-mediated processing of CXCL13 in situ.

To assess the effect of C-terminal truncation of CXCL13 on cellular responses, we compared CXCL13 and its cleavage product CXCL13[1–72] for their capacity to mobilize intracellular calcium in CXCR5-transfected Pre-B 300-19 cells. Both CXCL13 and CXCL13[1–72] induce a rapid, transient intracellular calcium rise (Fig. 5d, e). Analysis of internalization of CXCR5 by flow

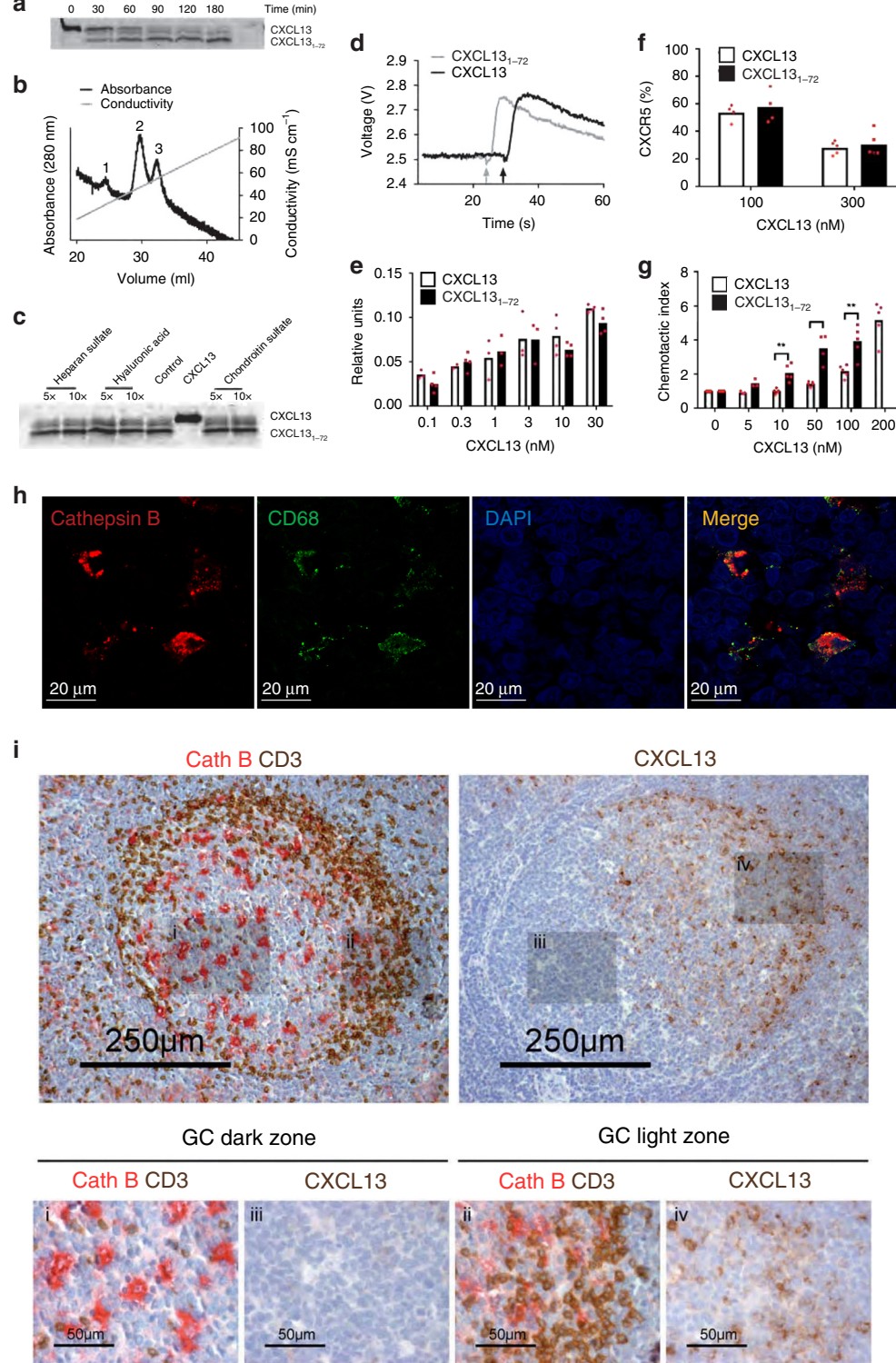

cytometry show that CXCL13 and CXCL13[1–72] are equally potent inducers of internalization at concentrations of 100 and 300 nM (42.6% vs. 46.6% and 69.43% vs. 71.7% internalization, respectively) (Fig. 5f). To determine if reduced binding of CXCL13[1–72] to heparin might also affect the chemoattractant activities for CXCR5+ cells, we studied in vitro migration of primary B cells expressing endogenous CXCR5 (Fig. 5g) and CXCR5-transfected Pre-B 300-19 (Supplementary Fig. 6) cells to CXCL13 and CXCL13[1–72]. In both assays CXCL13[1–72] displays greater potency than full-length CXCL13; for primary B

cells CXCL13[1–72]-induced migration at concentrations between 10 and 100 nM was significantly higher compared to full-length CXCL13 (Fig. 5g). Consistent with the 2D migration assays, CXCL13[1–72] induces more potent chemotaxis of CXCR5-transfected Pre-B 300-19 cells in a three-dimensional matrigel at lower ligand concentrations (Supplementary Fig. 6b).

To determine if Cath-B was expressed in the follicle, we performed IHC of tonsil tissue, with signal observed throughout the follicle, with highest expression colocalizing with CD68+ cells and some coexpression on CD35+ stromal cells (Fig. 5h, i).

**Fig. 5 Cathepsin B-mediated processing of CXCL13. a** 4 μM CXCL13 was incubated with 72 nM Cath-B for the indicated times at 37 °C. The cleavage products were separated by SDS-PAGE and stained with Coomassie blue. **b** C-terminal truncation of CXCL13 by Cath-B leads to decreased heparin binding. CXCL13 was incubated for 3 h with Cath-B, the reaction stopped, and the sample supplemented with intact CXCL13 and subsequently loaded on a HitrapTM heparin column. Proteins were eluted with a NaCl gradient of 0−1.0 M and absorbance measured at 280 nm. The three peaks were allocated as Cath-B (1), CXCL13[1–72] (2) and CXCL13 (3). **c** Processing of CXCL13 by Cath-B at pH 6.8 was unaffected by the presence of 5- or 10-fold (w/w) excess heparin sulfate, hyaluronic acid or chondroitin sulfate. **d** Representative $[Ca^{2+}]_i$ -dependent fluorescence changes in fura-2 loaded CXCR5-transfected Pre-B 300-19 cells induced by 30 nM CXCL13 or CXCL13[1–72]. **e** Dose response of calcium mobilization elicited by CXCL13 and CXCL13[1–72]. Relative units (mean ± SD) were calculated as described in "Methods". **f** CXCR5 surface expression after incubation of CXCR5-transfected Pre-B 300-19 cells with CXCL13 and CXCL13[1–72]. CXCR5 expression levels were quantified by flow cytometry analysis. Data (mean ± SD) from at least four independent experiments show the percentage of surface CXCR5 compared to control. **g** Primary human B-cell migration in response to intact and truncated CXCL13 was evaluated using 5 μm pore size Transwell filters. Data represent the percentage of migrated cells relative to the number of cells added to the Transwell filters. Values (mean ± SD) represent at least three independent experiments. For fig. 5g statistically significant differences (determined using a Student's t test) are indicated, *$p < 0.05$ and **$p < 0.01$. **h** Colocalization of Cath-B (red) and CD68 (green) signal in tonsil follicles. **h** Colocalization of Cath-B and CD68 staining in the B-follicle through immunohistochemistry analysis. **i** Analysis of Cath-B (Red), CD4+ T cells (brown) and CXCL13 in the B-cell follicle and germinal center reaction dark (subpanels i and iii) and light (subpanels ii and iv) zones. Source data are provided as a Source Data file.

Analysis of the Cath-B expression in the human GC reaction indicates a higher abundance of Cath-B-positive cells in the dark zone and CXCL13 producing stromal cells in the light zone (Fig. 5j). This is corroborated through analysis of tonsil tissue lysates by western blotting (Supplementary Fig. 7) and by data demonstrating that the in vitro culture medium of monocyte-derived macrophages is enzymatically active when assayed with the Cath-B-specific substrate Z-Arg-Arg-AMC. A small discernable effect of innate stimuli (LPS) on Cath-B function was observed, the significance of which during immune responses remained unclear (Supplementary Fig. 7).

To assess the in vivo importance of Cath-B in lymph node organization and function, we performed a detailed analysis of Cath-B ($Ctsb$)-deficient mice. Relative to wild type, $Ctsb^{-/-}$ lymph nodes are often visibly smaller (Fig. 6a), although there is no overall statistically significant decrease in the proportion of B cells in LNs (Fig. 6b). To determine the role of Cath-B in B-cell follicle formation staining of LNs was performed using antibodies specific for B-cell markers (CD19, B220), T cells (CD4), LN HEVs (PNAd) and stromal-cell subsets (Podoplanin, CD21/35). Strikingly, we found the morphology of follicles in $Ctsb^{-/-}$ lymph nodes is highly variable relative to WT. In many instances, we observed that follicles are not always discrete, but rather form a thin rim of B cells continued along the SCS and in many instances we observe a ring-like structure around the central T-cell zone (identified with immunoreactivity to CD4) (Fig. 6c).

This phenotype is suggestive of aberrant B-cell homing and follicle formation, possibly through defects in HEV formation or function. However, we could find no statistically significant difference in total B-cell numbers (Supplementary Fig. 11) and using immunohistochemistry (Meca-79) we did not observe defects in the HEV network (Fig. 6d). Additionally, to determine if B-cell homing is affected in Ctsb-deficient mice, CFSE-labeled CD45.1$^+$ B cells were transferred into either wild-type or $Ctsb^{-/-}$ recipients. No difference is found in B-cell homing into the LNs (Fig. 6e–g). In addition, confocal microscopy of LN sections shows that while CFSE$^+$ cells clearly overlap with B220$^+$ areas of WT animals, CFSE$^+$ cells are much more disperse and are found more frequently in B220 negative zones in $Ctsb^{-/-}$ mice. To corroborate these findings, we have performed RT-qPCR on whole LNs looking at a panel of genes relating to glycan synthesis and the formation of PNAd$^+$ HEV scaffolds ($Glycam1$, $Podxl$, $Cd34$, $Madcam1$, $FuctIV$, $FuctVII$), cellular adhesion ($Icam1$, $Vcam1$, $Pecam1$) and chemokines and their cognate receptors ($Cxcl13$, $Ccl19$, $Cxcr5$, $Ccr7$). With the exception of Podxl, we find no statistically significant difference in deltaCT values for each gene when comparing WT and $Ctsb^{-/-}$ mice (Fig. 6h and Supplementary Fig. 8). A small but nonsignificant decrease in

CXCL13 and CXCR5 was observed likely reflecting a failure in FDC network formation (Fig. 6g). These datasets suggest that CXCL13 can be solubilized by Cath-B, and that soluble CXCL13 gradients are essential for the formation of primary follicles within the LN. Taken in concert our data suggest that CXCL13 can exist in both immobilized and soluble forms, with availability fine-tuned by the reticular-cell microenvironment, and by the enzyme Cath-B.

## Discussion

Soluble factors are an essential means of communication between cells and their environment. In the context of the immune system, this cross-talk ensures that each B cell receives the appropriate signal at the appropriate time[5,45]. However, there is currently a lack of a well-accepted model to describe the spatial distribution of soluble factors in situ[25]. The data presented in this study highlights the importance of the tissue microenvironment in shaping gradients and raises the question of whether assuming free diffusion can provide sufficiently accurate theoretical models capable of generating novel predictions.

Using a modeling and simulation approach, we show that there is an underlying regulation to the spatial organization of CXCL13 at the cellular level, identifying a small-world network topology with regions of high connectivity and long-range connections between these cliques. These guidance structures are likely to promote trafficking of cognate B cells within the different niches of the B-cell microenvironment and the CR2-mediated delivery of large antigen from the subcapsular sinus to the B-cell zone reticular cell network by non-cognate B cells. Our data thus provide a unique insight into how the primary follicle is structurally organized to promote B-cell homeostasis and activation. We posit that the distinct topological properties of the reticular cell network with dense connectivity between cells are likely to create a labyrinth of single-cell niches, within which B cells scan for antigen. In future studies it would be of interest to assess whether the small-world properties of stromal cells in the primary follicle are maintained in the secondary follicle with a formed GC.

The implications of this cellular architecture also manifest at the molecular scale. By utilizing modeling and simulations in conjunction with imaging approaches, we propose a model whereby CXCL13 is largely immobile, with diffusion constrained by the localized tissue microenvironment. While our results indicate that heparin and heparan sulfate are important factors regulating the spatial distribution of CXCL13 it would also be of interest to know if other ECM components found in the follicle also contribute to CXCL13 immobilization. Importantly, our data suggest that immobilized CXCL13 is likely to form complex landscapes within tissues—a conceptual change in our

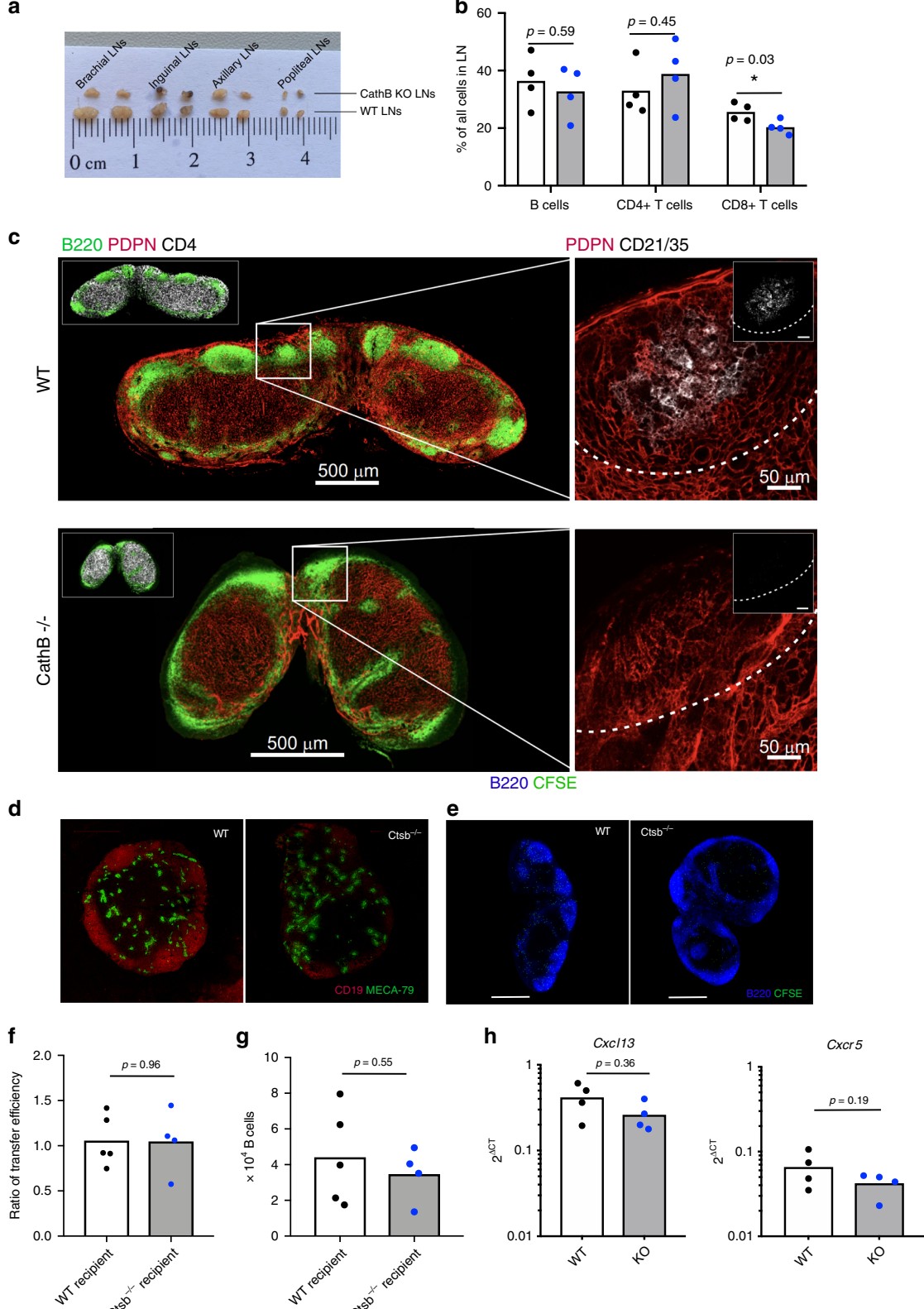

understanding of the form that gradients may take in vivo. Results from our multiobjective optimization emulation experiments suggest that this spatial profile is functionally important, promoting higher rates of scanning than homogeneous landscapes. These data are consistent with previous studies highlighting the importance of ECM components in modulating immune cell recruitment[15,18,21].

Interpretation of immobilized gradients may require proteolytic processing by Cath-B, yielding a truncated molecule capable of binding and signaling through CXCR5 but displaying reduced affinity for the ECM. Importantly, low concentrations of CXCL13 [1–72] were more potent then intact CXCL13 in attracting CXCR5-transfected Pre-B cells or primary B cells. Until recently, Cath-B in immune cells was regarded as a lysosomal enzyme

**Fig. 6 Cathepsin B-deficient mice have abnormal follicle architecture. a** Analysis of lymph node presence and morphology from WT and Ctsb[−/−] lymph nodes. **b** Percentage of B cells, CD4+ and CD8+ T cells in WT and Ctsb-deficient LNs determined using flow cytometry, with significance assessed using a Student's t test. **c** Staining of WT and Ctsb[−/−] LNs with anti-B220 (B cells), anti-Podoplanin (Stroma), anti-CD4 (T cells) and anti-CD21/35 (follicular dendritic cells). **d** Staining of WT and Ctsb[−/−] LNs for CD19 (B cells) and Meca-79 (PNAd+ HEVs). **e** Entry of CFSE transferred WT B cells into the LN parenchyma of either WT or Ctsb[−/−] recipient mice was assessed by confocal microscopy. **f** Ratio of LN entry of KO:WT B cells into either WT or Ctsb[−/−] recipients. To determine the relative efficiency of WT vs Ctsb[−/−] B cells to enter into WT or Ctsb[−/−] recipients, equal numbers of CSFE (ThermoFisher)-labeled KO cells and CMTMR (ThermoFisher)-labeled WT cells were transferred into corresponding recipient mice. The ratio of transferred B (B220[+]) cells KO:WT was calculated by taking into account the relative efficiency of CFSE and CMTMR labeled survival post transfer by calculating the ratio of WT CSFE:WT CMTMR transferred cells. **g** Quantification of migrated CSFE-positive B cells by flow cytometry. **h** Analysis of Cxcl13 and Cxcr mRNA expression from total LN from WT and Ctsb-deficient mice using RT-qPCR. For panels (**f**–**h**), significance was assessed using a Student's t test with p values provided for each comparison. Source data are provided as a Source Data file.

responsible for protein degradation, although cell membrane bound Cath-B has been shown to be functional in immune cells and can function across a range of pH values[46,47]. Our findings suggest extracellular occurrence and active secretion from both macrophages and reticular cells. Given that Cath-B activity is most potent at low pH values, and inflammation can lead to a decrease in tissue pH, it was interesting to note the increased secretion of Cath-B in the presence of LPS. However, it is unclear if this is an active release mechanism that occurs in vivo. It is possible that the initial influx of antigen triggers increased availability of CXCL13 at the subcapsular sinus, where antigen-presenting macrophages can then recruit both cognate B cells and non-cognate B cells to facilitate GC seeding and antigen deposition on the B-cell zone reticular cell network. Strikingly, follicular architecture in Ctsb-deficient mice bears a strong resemblance to the phenotype observed in lymphoid tissues of CXCL13-deficient mice[7], and in the spleens of CXCR5-deficient mice[6]. This is consistent with a model where soluble CXCL13 drives chemotactic homing behaviors while immobilized CXCL13 promotes haptokinetic scanning within the follicle, as has been demonstrated for CCL21[15]. In future studies it would be interesting to assess the validity of this model and to assess whether perturbing Cath-B-mediated regulation of CXCL13 in vivo can alter the onset and efficacy of affinity maturation, and whether other enzymes are involved in CXCL13 processing.

Engineering approaches often draw inspiration from natural systems to solve complex design problems; however, they can reciprocally influence our understanding of the immune system, providing a quantitative framework from which to understand the spatial distribution of morphogens. Using an ensemble of different techniques, we were able to consolidate several disparate datasets and through simulation-based experimentation have generated insights that informed subsequent experimental work. Specifically, we have highlighted the use of data-driven machine learning and evolutionary computational approaches to expedite the translation of simulator-derived insights into a better understanding of the design, organization, dynamics, and function of complex biological systems.

In conclusion, our data suggest that CXCL13 can exist in both immobilized and soluble forms, with the precise mode of availability dependent on enzymatic processing by Cathepsin B. This provides a significant update in our conceptual understanding of how homeostatic chemotactic gradients arise and form functional gradients in complex tissues.

## Methods

**Enzymatic treatment of tonsil sections**. Frozen lymph node or tonsil sections on polylysine slides were incubated at room temperature for 30 min. A circle was drawn around each section using a wax ImmEdge pen (Vector Laboratories), the sections were then hydrated with PBS for 5 min and incubated with 150 nM recombinant Cath-B (Sigma-Aldrich) for 3 h at 37 °C or with 10 U heparinase II (Sigma-Aldrich) for 1 h at 17 °C. Slides were washed in PBS and then processed for immunohistochemistry (as described below) with no fixative. All samples were

ethically approved and informed consent was obtained from all participants. Tonsils were collected under NRES REC 12/NE/0360-approved study (IRAS: 114771) to M.C.C. Hepatic lymph nodes were collected during multiorgan donation procedures, after approval by the Medical Ethical committee of the Erasmus MC (MEC-2014-060) by WGP.

**Immunohistochemistry and immunofluorescence**. Frozen lymph node or tonsil sections on polylysine slides were incubated at room temperature for 30 min, fixed in acetone or 4% paraformaldehyde (PFA) and then washed in PBS for 15 min in total with changes of PBS every 5 min. Sections were incubated in a blocking buffer of PBS and 5% serum (the serum of the host secondary antibody was raised in) at room temperature for 1 h at room temperature. After blocking, sections were incubated in the primary antibody mix, made up in blocking buffer for 1 h at RT. The slides were then washed, and secondary antibody incubation was performed (if necessary). For experiments where exogenous CXCL13[AF647] was used to measure binding to tissue, incubation of unfixed tissue sections with 500 nM CXCL13[AF647] for 1 h at RT instead of the secondary antibody-staining step. Samples were washed for 5 min in PBS. A drop of Prolong gold (Invitrogen) was added to each section, and then a No. 1.5 glass coverslip (Fisher) mounted on top. The slides were incubated overnight at 4 °C, the next day slides were sealed using nail varnish and stored at 4 °C. Immunofluorescent-stained sections were imaged using the Zeiss LSM 880 confocal microscope. Samples were excited with 405, 488, 561, and 633 nm lasers. Image acquisition was performed using the ×63 oil objective. Tile scans and Z stacks were performed to image large tissue sections at high resolution. For imaging of chemokine gradients, we used the Airyscan module to increase spatial resolution beyond the diffraction limit of light. A list of commercial antibodies used in this study are available in Supplementary Table 2.

For immunohistochemistry on human tonsil sections, specimens were fixed in 10% buffered formalin, embedded in paraffin and cut into 4-μm cross-sections for immunostaining. Deparaffinized and rehydrated sections were boiled at 95 °C for 30 min in target retrieval solution (S1699 DAKO) and then treated with peroxidase blocking reagent (S2001, DAKO) when needed, and protein block serum-free (X0909, DAKO). Sections were incubated overnight at room temperature with anti-CD3 at 5 μg/ml, anti-Cath-B at 0.12 μg/ml and anti-CXCL13 at 1 μg/ml. Next biotinylated anti-mouse IgG, anti-rabbit IgG, or anti-goat IgG were used at 2 μg/ml and applied for 30 min at room temperature. Slides were washed and incubated with StreptABComplex (K0377, K0391, DAKO). Double-staining for CD3 and Cath-B was performed in two steps; slides were blocked with 3 μg/ml rabbit IgG (X0936, DAKO) after incubation with anti-CD3. For CXCL13 single staining in immunohistochemistry, after anti-CXCL13 antibody, sections were incubated with MACH1 with primary antibodies, the sections were incubated with corresponding secondary antibodies according to the manufacturer's instructions. Sections were developed with either DAB or New Fuchsin and nuclei counterstained with hematoxylin. For immunofluorescence stainings, after incubation with primary antibodies, the sections were incubated with corresponding secondary antibodies from Alexa for 30 min and then nuclei counterstained with DAPI.

Mouse lymph node frozen sections (8 μm) from Ctsb[−/−] and controls were hydrated and washed using PBS; each wash step was 5 min, repeated three times. Sections were incubated in blocking buffer (PBS 5% goat serum) at room temperature for 5 min. Following blocking sections were incubated in a primary antibody-staining mix, made up in blocking buffer, for 1 h at room temperature. Slides were washed, then incubated in secondary antibody-staining mix, made up in blocking buffer, for 1 h at room temperature. Following a final wash ProLong Gold (Invitrogen) was added to each section, then a No. 1.5 glass coverslip mounted, slides were incubated overnight at 4 °C and sealed with nail varnish. The antibodies used in staining mixes were: MECA-79 Alexa488 (Nanotools (Custom Product), 1 in 200 dilution); PDPN Alexa 594 (Biolegend (8.1.1) (Cat. 127414); B220 Alexa488 (Biolegend (RA-6B2) (Cat. 103225), 1 in 200 dilution); CD4 Alexa647 (Biolegend (RM4-5) (Cat. 100516), 1 in 200 dilution); CD21/35 Alexa647 (Biolegend (7E9) (Cat. 123424) 1 in 200 dilution); and CD19 Alexa647 (Biolegend (6D5) (Cat. 11512), 1 in 200 dilution). All experiments involving mice conformed to the ethical principles and guidelines approved by the University of York

Institutional and Animal Care Use Committee in accordance with the European Union regulations and performed under a United Kingdom Home Office License.

**Reticular cell topology**. Topological analysis was performed using the methodology as previously described[48]. 3D images (approx. $450 \times 450 \times 35 \mu m$) of lymph nodes from Cxcl13-EYFP mice were obtained by laser scanning confocal microscopy. Experiments were performed in accordance with federal and cantonal guidelines (Tierschutzgesetz) under permission numbers SG10/16, SG07/16 and SG05/15 following review and approval by the Cantonal Veterinary Office (St. Gallen, Switzerland). The topological mapping of follicular stromal cell network structure was created as an undirected unweighted graph by defining nodes as the $EYFP^+RFP^+$ follicular stromal cells and edges as physical connections between neighboring nodes. The network edges in 3D Z-stack images were annotated using the Measuring Tool in Imaris (Bitplane) such that a straight line is demarcated between adjacent stromal cells that are connected by a cellular protrusion or smaller branching process with no other cell body directly blocking this connection. Analysis of key topological parameters (described in Supplementary Table 1) was performed using the iGraph package in R. These parameters enable the assessment whether the network has small-world properties as has been reported for T-cell zone FRC networks in lymph nodes[32]. Although many additional topological and structural metrics exist, the metrics proposed in this study are sufficient to perform a basic characterization of the follicle network, while also providing quantitative data to inform the algorithmic reconstruction of an in silico stromal network model.

**Quantifying the spatial autocorrelation of fluorescence**. To quantify the spatial autocorrelation of fluorescence intensity, 2D confocal images were acquired on a Zeiss LSM 880 confocal microscope with the same laser settings and post processing for each sample. Processed .png files were then analyzed in R using custom scripts. Briefly, this analysis involved discretizing the image into $14.44 \mu m^2$ bins and calculating the spatial correlogram using the correlog function from the ncf package. Spatial autocorrelation is quantified using Moran's $I$ statistic with significance assessed through permutation testing[49,50].

**Super-resolution imaging**. Frozen tonsils sections on polylysine slides were incubated at room temperature for 30 min. Samples were hydrated in PBS for 5 min, then left to dry and circles were drawn around each section with a wax ImmEdge pen (Vector Laboratories). Sections were incubated in a blocking buffer of PBS + 5% goat serum (Sigma) at room temperature for 1 h. After blocking, sections were incubated in primary antibody mix (anti-B220 FITC, eBioscience) made up in 1:200 blocking buffer for 1 h at room temperature. Samples were washed with PBS for $3 \times 5$ min and 30 nM of CXCL13-AF-647 were added to the slides. Slides were left to incubate overnight at 4 °C after which slides were washed for 30 s in PBS and a No. 1.5 glass coverslip (Fisher) mounted on top.

Bespoke fluorescence microscopy was performed on an inverted microscope (Nikon Eclipse Ti-S) with a ×100 NA 1.49 Nikon oil immersion lens and illumination from a supercontinuum laser (Fianium SC-400-6, Fianium Ltd.), controlled with an acousto-optical tunable filter to produce a narrow-field excitation light centered on 619 nm[51,52]. The use of narrow-field imaging permits fluorescent excitation at distance of a few hundred nanometers above the coverslip thus mitigating some of the boundary effects that may be encountered using total internal fluorescence microscopy where only a thin section directly above the coverslip is excited[53]. A 633-nm dichroic mirror and 647-nm long-pass emission filter were used to filter the appropriate wavelengths of light emitted from the fluorescence images. Images were recorded on an emCCD camera (860 iXon$^+$, Andor Technology Ltd) cooled to −80 °C. $128 \times 128$ pixel images were acquired for 1000 frames with 1.98-ms exposure times. The camera was in frame transfer mode with the resulting frame rate being 513 Hz. The electron-multiplier gain was set to 300. The kinetic series were saved as TIFF format files (.tiff). When imaging in tissue, sections were stained with an anti-B220 (1 in 200 dilution) antibody conjugated to FITC. Samples were imaged at low (1.2 µm/pixel) magnification with green illumination (470 nm) to determine the location of the B-cell follicles, before switching to high (120 nm/pixel) magnification and red illumination to image chemokines in these areas.

The analysis of the kinetic series was done in bespoke Matlab software, namely ADEMS code[52], which enabled objective single-molecule detection and tracking to within 40-nm spatial precision, utilizing a combination of iterative Gaussian masking and local background subtraction to calculate sub-pixel precise estimates for the intensity centroid of each candidate fluorescent dye in the image with edge-preserving filtration of intensity data and Fourier spectral analysis to confirm detection of single dye molecules[54–57]. The code was first performed on simulated kinetic series that mimicked the signal and noise landscape of real image data. The parameter settings such as values for the signal-to-noise ratio and the Gaussian mask size of ADEMS code were set so that the code accurately identified the signals in the simulated data. These parameters were then used in the code for the identification of single fluorescent signals in real data. From these fluorescent spots ADEMS code then produced trajectories of fluorophores that last five or more consecutive frames to allow the calculation of microscopic diffusion coefficients as the gradient of a linear fit to the first four positions in each track[58,59]. These

coefficients were plotted in histograms with integer bin sizes for easy comparison between the experiment and the control groups.

**Emulator development**. As an agent-based model, a number of high-level properties emerge from the simulator due to aggregated interactions between agents and their environment[60,61]. To learn the complex relationship between parameter inputs and emergent agent behaviors, we employ a supervised machine-learning approach. Supervised learning involves generating a dataset of inputs ($x$) and outputs ($y$) and then teaching an algorithm to approximate a mapping function between the two. With a sufficiently accurate mapping function, it is then possible to predict $y$ for a set of unobserved values of $x$.

The training dataset for emulator development was obtained using Latin hypercube sampling[62], with 3000 parameter sets. Each set was executed 100 times to mitigate aleatory uncertainty, and median responses calculated to summarize simulator performance under those conditions.

To map the complex relationship between parameter inputs and the emergent properties of the model, we train an ANN using the SPARTAN[63] package in R. ANNs are a technique inspired by the neuronal circuits in the brain, with computations structured in terms of an interconnected group of artificial neurons organized in layers. In this scheme parameter inputs are passed into the network and iteratively processed by a number of hidden layers. Within each hidden layer the sum of products of inputs and their corresponding weights are passed through a sigmoidal activation function that is fed as inputs into the next layer. This process is repeated until the output layer is reached and we have a prediction for the output values. During the learning phase, the weighting of connections between neurons is adjusted in such a way that the network can convert a set of inputs (simulation parameters) into a set of desired outputs (simulation responses)[64].

A key technical consideration when developing neural networks is how to evaluate predictive power. Testing predictive performance on the training data is not useful as it can lead to overfitting, whereby the network is poor at predicting previously unobserved data. To solve this problem, a proportion of the dataset is omitted from the training dataset and used to validate algorithm performance. To evaluate the predictive power of the emulator, we partition the LHC dataset into training (75%), testing (15%), and validation (10%) datasets. Partitioning the data incurs a cost however, as we reduce the number of samples used for training the model. In addition, the data used to train the model, even if not used in the evaluation process, can have a significant impact on predictive performance. To address these issues, we perform a procedure known as $k$-folds cross-validation. In this scheme the data are partitioned into $k$-folds and the algorithm learns the mapping between inputs and outputs using $k - 1$ folds as training data with validation performed on the remaining part of the data. This process is repeated until each fold is used as the test set with overall performance taking as the average for each fold. To develop our ANN, we generate multiple neural network structures with different number of hidden layers and nodes within each layer (so-called hyperparameters) but fixed input and output layers (one node for each distinct input and output respectively). The accuracy of each putative network was quantified using the root mean squared error between the predicted cell behavior responses and those obtained by the simulator. Using this approach an ANN was developed for each simulation output metric with network structures presented in Supplementary Fig. 9.

**Multiobjective optimization**. Multiobjective optimization analysis was performed using the nondominated sorting genetic algorithm II (NSGA-II), a multiobjective genetic algorithm[39]. This analysis was performed in R using the package mco v15.0. The four objectives to be optimized by the algorithm were to: minimize the root mean squared error between emulator and simulator responses for cell speed, meandering index and motility coefficient, and maximize scanning rates.

**Preparation of recombinant Cxcl13**. Preparation of recombinant CXCL13, full-length and 1–72 form was prepared as described previously[65]. CXCL13 labeled with the fluorescent tag AF647 was purchased from Almac.

**CXCL13-processing by Cath-B**. Synthetic human or mouse CXCL13[66] was incubated with purified human liver Cath-B (Athens) (for mice CXCL13 was incubated with recombinant Cath-B purchased from R&D Systems) at 37 °C in Dulbecco's PBS (DPBS, Invitrogen) pH 6.8 containing 4 mM Ethylenediaminetetraacetic acid (EDTA) and 2 mM L-cysteine. The reaction was stopped by boiling the samples at 95 °C for 5 min. The chemokine cleavage products were separated by Tris-Tricine SDS-PAGE and stained with Coomassie blue. Enzymes were activated as per the manufacturer's instructions.

**Interaction of CXCL13 with glycosaminoglycans**. CXCL13 and CXCL13[1–72] binding to heparin was characterized by loading respective chemokine samples on a 1 ml Hitrap$^{TM}$ heparin column (GE Healthcare). Bound CXCL13 and CXCL13 [1–72] were eluted using a linear gradient of 0−1.0 M NaCl in 10 mM potassium phosphate, pH 7.5 over 30 min at a flow rate of 1 ml/min and monitored by absorbance at 280 nm on a DuoFlow system (Bio-Rad). The impact of soluble GAGs on CXCL13 processing by Cath-B was determined by performing CXCL13

cleavage experiments in the presence of hyaluronic acid, heparan sulfate or chondroitin sulfate (Sigma).

**Intracellular calcium mobilization**. CXCL13-induced changes in cytosolic-free $Ca^{2+}$-concentration $[Ca^{2+}]_i$ were measured in CXCR5-transfected mouse Pre-B 300-19 cells[24]. Cells were loaded with 0.2 nmol of fura 2-AM per $10^6$ cells for 20 min at 37 °C in a buffer containing 136 mM NaCl, 4.8 mM KCl, 1 mM $CaCl_2$, 1 mg/ ml glucose and 20 mM Hepes, pH 7.4 (MSB). After centrifugation, fura-2-loaded cells were resuspended in MSB, stimulated at 37 °C with the indicated concentrations of intact or truncated CXCL13, and the $[Ca^{2+}]_i$-related fluorescence changes were recorded as previously described[67]. Relative units are calculated as the ratio of the fluorescence signal after chemokine stimulation and a calibration signal.

**CXCR5 internalization**. $3 \times 10^5$ CXCR5-transfected Pre-B 300-19 cells were washed with DPBS and incubated for 45 min at 37 or 4 °C (control) in 50 µl DPBS containing 2% PPL (human albumin, CSL Behring) and the indicated concentrations of intact or truncated CXCL13. Cells were then washed with ice-cold DPBS supplemented with 1% BSA (fraction V, Applichem) and 0.04% sodium azide and blocked with 3 mg/ml Vivaglobulin (CSL Behring) for 12 min at 4 °C before incubation with anti-human CXCR5 antibody (1:40) or isotype control (1:40). Surface receptor expression was evaluated by flow cytometry.

**Cell migration**. Two-dimensional chemotaxis assays with human B cells and CXCR5-transfected Pre-B 300-19 cells were carried out in 5-µm pore size Transwell plates (Costar). Cells were washed and resuspended at $5 \times 10^6$ cells/ml in RPMI containing 10% FBS (Invitrogen), L-glutamate, sodium pyruvate and 2-mercaptoethanol. Chemokines were diluted in the same buffer and added to the wells. Filter inserts were then placed in the wells and the assay was started by adding 100 µl of cell suspension into the filter inserts. After 2 h at 37 °C and 5% $CO_2$, the filter inserts were removed, and the migrated cells counted by flow cytometry (FACSCalibur Becton Dickinson) for 30 s using a high flow preset. Assays were carried out in duplicates and tests from different days were standardized by measuring PKH26 reference microbeads (Sigma) under the same conditions. Cell migration in a 3D setup was assessed as described previously[68,69]. Briefly, $5 \times 10^4$ CXCR5-transfected pre-B 300-19 cells in 100 µl RPMI 1640 medium supplemented with 10% fetal calf serum and 0.1% $\beta$-mercaptoethanol were pre-mixed at 4 °C with growth-factor reduced Matrigel (Corning, BD Biosciences #356231) to a final Matrigel concentration of 300 µg/ml and seeded to the upper well of a 24-well Transwell$^{TM}$ System and polycarbonate filters with a pore size of 5 µm (Corning Costar). Matrix was allowed to polymerize for 2 h at 37 °C/5% $CO_2$. Cells were subsequently allowed to migrate through the Matrix for 3.5 h towards the lower well containing graded concentrations of the chemokines. The numbers of input and migrated cells were determined by flow cytometry (LSRII, BD Biosciences).

**Quantifying lymph node cellularity**. Accucheck counting (Invitrogen) beads were used to calculate total cellularity of murine popliteal lymph nodes. Following antibody staining, pellets were resuspended in 100 µl FACS wash. One hundred microliters of counting beads were mixed for 1 min to ensure they were evenly resuspended before running on the flow cytometer. To ensure accuracy the beads are made up of two types of beads that differ in their fluorescent intensity; for accurate readings, the two populations should be present at approximately 50:50 ratio. To calculate the absolute cell number, the following calculation was then made:

$$\text{Number of cells per µl} = \frac{\text{Number of events (beads)}}{\text{Number of events}} \times \text{Number of beads per µl},$$

where the number of beads per µl was provided by the supplier and varied with each batch. The total cellularity of the lymph node could be calculated using the values of cells per µl and final volume of FACS wash that contained the cells.

**Real-time quantitative PCR**. RNA was extracted from whole LNs using RLT buffer (Qiagen), the lysates were stored at −20 °C overnight. RNA extraction was performed using an RNeasy mini kit (Qiagen). Quantity and quality of RNA was measured using a NanoDrop spectrophotometer. Samples were stored at −80 °C. cDNA synthesis was performed using the high-capacity cDNA Reverse Transcription Kit (Applied Biosystems) and a thermo cycler PCR machine. cDNA samples were stored at −20 °C. The following Taqman probes (ThermoFisher cat. 4331182) were utilized: CCL19 (Mm00839966_g1); CXCL13 (Mm04214185_s1); GlyCAM-1 (Mm00801716_m1); Podxl (Mm00449829_m1); CD34 (Mm00519283_m1); MADCAM (Mm00522088_m1); FucT IV (Mm00487448_s1); FucT VII (Mm04242850_m1); VCAM (Mm01320970_m1); PECAM (Mm01242576_m1); ICAM (Mm00516023); CXCR5 (Mm00432086_m1); CCR7 (Mm99999130_s1).

**B-cell in vivo homing assays**. B220$^+$ B cells were isolated from CD45.1 congenic mice, and stained with anti-B220, sorted on a S3 Bio-Rad cell sorter and labeled with CFSE for 10 min in serum-free medium. Labeled cells were washed in

complete medium prior to resuspension in PBS, $10^7$ B220$^+$ CFSE labeled were transferred intravenous into either Ctsb$^{-/-}$ or wild-type recipients. Twenty-four hours post transfer, LNs were isolated and 8 mm frozen sections cut. Sections were counterstained with anti-B220 Alexa647 (Biolegend) and imaged on a Zeiss 810 confocal microscope. Experiments were performed in accordance with federal and cantonal guidelines (Tierschutzgesetz) under permission numbers SG10/16, SG07/16, and SG05/15 following review and approval by the Cantonal Veterinary Office (St. Gallen, Switzerland). To determine the relative efficiency of WT vs. Ctsb$^{-/-}$ B cells to enter into WT or Ctsb$^{-/-}$ recipient mice, equal numbers of CSFE (ThermoFisher)-labeled KO cells and CMTMR (ThermoFisher)-labeled WT cells were transferred into the corresponding recipient mice. The absolute number of B cells was determined by multiplying absolute cell counts from individual matched inguinal LNs using CASY with flow cytometry analysis of isolate lymphocytes with CD19-APC (Biolegend) and CD3eBrilliantViolet (Biolegend) on a FortessaX20 (BD). The ratio of transferred B (B220+) cells KO:WT was calculated in both LN and spleen of the different recipient mice taking into account the relative efficiency of CFSE and CMTMR labeled survival post transfer by calculating the ratio of WT CSFE:WT CMTMR transferred cells (Cell number × % B cells of CFSE$^+$ or CMTMR$^+$ populations). The gating strategy for flow cytometry analysis is shown in Supplementary Fig. 12. This methodology removed the effect of CMTMR nonspecifically affecting the efficiency of dye labeled lymphocyte survival post transfer.

**Reporting summary**. Further information on research design is available in the Nature Research Reporting Summary linked to this article.

## Data availability
Source Data are provided in Zip folder. All raw datasets (1.5 GB zip file) that support the findings of this study are available from the corresponding author upon reasonable request.

## Code availability
A brief overview of the simulator platform is presented in Supplementary Note 3. A full description of model simulator design, development and validation, as well as associated source code, is available from https://www.kennedy.ox.ac.uk/technologies/resources/cxcl13sim.

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

## Acknowledgements

We wish to thank Antal Rot, Paul Kaye, Dimitris Lagos and members of the York Computational Immunology Laboratory for advice and reagents, the York Teaching Hospital

NHS Foundation Trust R&D Department for invaluable assistance with sample collection protocol and Imaging & Cytometry Laboratory staff for technical input. Work was funded by the Swiss National Science Foundation Grant (310030_163336) to M.T. and Swiss National Science Foundation Grants 159188 and 166500 to B.L. M.C.L. was supported by the Biological Physical Sciences Institute (BPSI), Medical Research Council grants MR/K01580X/1 (M.C.L.), MC_PC_15073 (M.C.C., M.C.L. and Z.Z.) and BBSRC grants BB/N006453/1 and BB/R001235/1 (M.C.L.). M.W. was supported by the Bernische Krebsliga to M.W. and S.A., by the Swiss European Union FP6 (INNOCHEM, LSHB-CT-2005-518167), the Swiss National Science Foundation (143718 to M.U.) and the San Salvatore Foundation to M.U., the Swiss National Science Foundation (169936) to D.F.L. D.V. was supported by the MD/Ph.D. scholarship from the Swiss National Science Foundation and the Max Cloëtta Foundation (313600-115688). B.M. was supported by INSERM U1151. K.A. was supported by Wellcome Trust Centre for Future Health grant (204829), J.T. by EPSRC grant EP/K040820/1. J.C., J.T. and M.C.C. were funded by Wellcome Trust (Computational Approaches in Translational Science WT0905024MA, HFSP (RGP0006/2009 T.C. and M.C.C.) and Medical Research Grants MR/K021125/1 and G0601156. M.C.C. is funded by the Kennedy Trust.

## Author contributions

J.C., M.N. designed and performed the experiments, analyzed and interpreted the data and wrote the manuscript; S.A., N.B.P., Z.Z., L.O., U.M., J.C., H.M., K.A., A.T., S.J., E.T., D.V., M.H., M.U., C.J.L., A.C., P.O.T., R.P., W.G.P., and D.F.L. performed experiments, analyzed data, provided key reagents, intellectual input, and technical assistance; M.T., T.C., B.M., J.V.S., M.W. designed experiments and analyzed data, M.W., M.C.L., J.T., B.L., and M.C.C. designed experiments, analyzed and interpreted results, coordinated the research and wrote the paper.

## Competing interests

The authors declare no competing interests.
