## [Peer Review File · Nature Communications]

Reviewers' comments:

Reviewer #1 (Leukocyte trafficking, imaging) (Remarks to the Author):

In this paper the authors suggest that reticular cells of the lymph node B cell zone generate microenvironments that shape CXCL13 gradients. By using computational approaches, they suggest that CXCL13+ follicular reticular cells form a small-world network of guidance structures and predict that immobilized gradients of CXCL13 support B cell trafficking. Primary B cell follicles fail to form in mice lacking cathepsin B potentially due to a lack in soluble CXCL13.

The computational approaches and mathematical modeling are original in the present context. The topological analysis of follicle networks using systems biology approaches is robust and includes an appropriate control (random networks). The optimization analysis and pattern recognition parts are appropriate and convincing. The theoretical analysis predicts that immobilized CXCL13 gradients are critical for B cell trafficking within the follicle, and the immunofluorescence studies suggest that the lack of cathepsin B interferes with B cell migration and follicle formation. However, although the idea behind the study and the methodological approaches are original, the study lacks appropriate in vivo data to support the authors' claims.

The present study raises major concerns that need to be addressed.

1. It is unclear how many cells (network nodes) were analyzed for each lymph node and cell type (FDC and RC) and how many lymph nodes were used for each mouse for the data shown in Figure 1.
2. Figure 1.E. Were the shortest path results for the global follicular network compared to a random network? Are these results statistically significant?
3. The authors do not explain how the small world configuration of CXCL13+ cells may promote both complement-mediated trafficking of antigen and the migration of cognate B cells as stated in the Results section (page 8).
4. To assess the in vivo importance of cathepsin B, the authors performed a histological analysis of lymph nodes from *Ctsb* deficient mice, suggesting a defect of B-cell homing. However, the authors should provide clear in vivo demonstration that cathepsin B deficiency affects intranodal B cell migration.
5. Cathepsin B deficiency may also affect T cell migration and lymph node development. How can the authors exclude this aspect?
6. Do cathepsin B deficient mice have a defect in B cell adhesion in high endothelial venules, potentially explaining the reduction of LN size?
7. Are the results shown in Figure 2D statistically significant?
8. The data from Figure 5J suggest some overlap between CD35+ cells and expression of cathepsin B.
9. Figure S9a mentioned in the text is missing.
10. Some histograms are missing from Figure N4.3.
11. The authors should check figure legends 3 and 4 (these two fig. legends seem inverted).
12. The authors mention in the discussion that "Our data thus provides a unique insight into how the primary follicle is structurally organized to promote B-cell homeostasis and activation." However, no convincing in vivo functional data are provided to support this claim.

Minor points:

- Page 13 line 306: "figure 6k" should be replaced with "figure 5k".
- Line 305: "a phenotype suggestive of defective B-cell homing" is repeated two times.
- The significance bars need to be checked in Figure 6S.

Reviewer #2 (Lymphocyte trafficking, endothelial cell)(Remarks to the Author):

Cosgrove et al. have used mathematical modelling and biological analyses in studying CXCL13 chemokine gradients in B-cell follicles of lymph nodes. They report that CXCL13 binds to heparin/heparin sulfate- containing molecules in the extracellular matrix and that protease cathepsin B can cleave this chemokine with a possible concomitant increase in potency. Finally, they describe altered B-cell follicle morphology in cathepsinB-deficient mice.

Major concerns:

1. The manuscript reproduces many findings already firmly established in the field (e.g. the production of CXCL13 by follicular dendritic and other reticular cell types and CXCL13 binding to heparin sulphate). In general the manuscript does not significantly add to the understanding of the standard "production, diffusion, immobilization, consumption"-model of chemokine function.

2. Heparin-binding of CXCL13: If CXCL13 binds to heparin in follicles, how does it bind to much brighter extrafollicular heparin sulfate positive structures (possibly vessels?) in tissue sections (Fig 3A; is the numbering of Figs 3 and 4 swapped?). If the quantification of the binding of Alexa-labeled CXCL13 to tonsil tissues (Fig. 3 B) after enzymatic treatments is done from images like Fig. 3A (two lower rows) it will be biased since the extrafollicular areas are excluded. How does the Alexa-labeling affect the binding?

3. The role of cathepsin-B (a well-known lysosomal enzyme) in proteolytic cleavage of CXCL13 is interesting, but very preliminary and highly speculative at this stage. The rationale for choosing cathepsin B as a putative candidate for CXCL13 degradation seems to be missing. The analysis of cathepsin B expression in human tissues is not very convincing (the use of polyclonal antibody in stainings would require much more stringent specificity controls; the signal in Western blotting can come from any cell type/anatomical compartment; finding of cathepsin B in supernatant of in vitro cultured monocytes/macrophages is not directly relevant to lymphoid tissues). In the absence of any evident alterations in signaling (Fig 5D,F), the mechanisms and in vivo relevance of greater potency (Fig 5E,G) of cleaved CXCL13 observed in vitro chemotaxis assays remain unclear. In fact, the authors do not seem to provide any evidence that such cleaved form exists in vivo in humans.

The analysis of Cathepsin B-deficient mice is very preliminary, and from the data shown there is no way of claiming that the observed B-follicle phenotype would be linked to CXCL13 processing. For instance, the size difference of lymph nodes, and the possible alterations in other cell types, in follicular morphology and chemokine/chemokine receptor expression should be analyzed in much more detail and quantitatively. Mouse and human CXCL13 only share about 65% similarity in the protein sequence. Does cathepsin B cleave mouse CXCL13 even in vitro? Is there any difference in the levels of full-length and truncated CXCL13 in the lymph nodes of these mice? Are there any defects in B-cell homing to lymph nodes or in parenchymal B-cell migration to follicles in vivo in cathepsin B knockouts?

4. Much of the data are from a single patient (Figs. 3, 4).

Reviewer #3 (Computer modelling of immune system)(Remarks to the Author):

The study addresses a fundamental issue of how the functioning of B cells in lymph nodes is organized structurally and is regulated biochemically. The authors demonstrate that both the soluble and immobilized of the chemokine CXCL13 play have key roles which depend on the context. The

immobilized CXCL13 gradients in B cell follicles are characterized by spatially complex patterns. They are built up by the reticular cell network to provide an 'informed' guidance structure for targeted B cell trafficking within complex cellular environments. The paper introduces a number of key advancements in our understanding the structural (topological) organization of the follicular reticular cell network, the mechanisms underlying the switch between haptotaxis and chemotaxis of B cells in the follicles, and the homeostasis parameters determining the availability of immobilized versus soluble forms of CXCL13. An important central component of this interdisciplinary study is a novel efficient methodology of using elaborate multiscale computational modelling to gain a predictive understanding of the system behavior.

1. Page 8-9: The Pareto optimal solutions need to be explained and illustrated either in the main text or in supplementary notes.
2. Application of the artificial neural network-based (black-box) emulator of the multiscale (mechanistic) model rises the issue of whether its sensitivity to the parameter variations is the similar to that of the mechanistic model itself. This needs to be discussed.
3. What is the statistical framework used to assimilate the empirical data using the mechanistic model? Were the uncertainty intervals evaluated?
4. Page 21: Training of artificial neural network (ANN): What is the rationale behind the selecting a given number of the hidden layers and the number of their elements. Four different structures are shown in Figure S7. Were the information-theoretic criteria used to rank the ANNs? Please, could you elaborate more on this.
5. Supplementary Note 1, page 8: the formulas for Global clustering coefficient, Average local clustering coefficient and in part, for Sigma factor are not consistent with the text and notations around them.
6. Figure 1E, length units are missing.
7. Figure 2E, Decay rate: What is the implication of the calibrated values of the parameter to be located at the edge of the curve corresponding to Pareto optimal solutions?
8. Figure 3 and Figure 4 need to be reordered to fit the legends.
9. Table N4.1: "...and parameters were removed where possible." This is not clear.
10. Figure N4.4.: A,B,C are missing in the legend.
11. Figure N4.3: There are empty spaces instead of graphs.
12. Supplementary Note 4, page 19: "...the number of cells in the target location." This is not clear, i.e. the concentration or the population size are implied.
13. Supplementary Note 4, page 20: " $\ln(\alpha)$ " is not explained. Is it a function?

Reviewer #1

- 1. It is unclear how many cells (network nodes) were analyzed for each lymph node and cell type (FDC and RC) and how many lymph nodes were used for each mouse for the data shown in Figure 1.**
- 2. Figure 1.E. were the shortest path results for the global follicular network compared to a random network? Are these results statistically significant?**
- 3. The authors do not explain how the small world configuration of CXCL13+ cells may promote both complement-mediated trafficking of antigen and the migration of cognate B cells as stated in the Results section (page 8).**

The data regarding the number of network nodes per lymph node area, as well as other network parameters, are indicated in Supplementary Table S1. The number of mice used for the topological analysis is indicated in Figure 1 legend. The results displayed in Figure 1E indicate the distribution of the topological shortest path lengths for the global follicular network. We have included now the comparison of the average shortest path lengths of the B cell follicle networks (n=4 mice) and equivalent random networks (n=4, averaged across 1000 simulations per mouse) in Figure 1F. Comparing the average shortest path lengths by means of an unpaired Student's t-test confirms statistically significant differences ($p=0.0029$).

Indeed, the reviewer is correct, we did not sufficiently explain how the small-world topology impacts trafficking of antigen in B cell follicles. It has been shown in many previous topological studies that the small-world configuration optimizes information transfer across the network by generating high local neighborhood connectivity (clustering coefficient) and linking shortcuts to more distant nodes (shortest path length). We refer the reviewer to reference 33 in the manuscript: *Watts, D. J. & Strogatz, S. H.*

Collective dynamics of 'small-world' networks. Nature (1998). Thus we have incorporated in Figure 1F the average shortest path lengths compared to random networks, to better illustrate this point. We also direct the reviewer to Table S1, where a 10-fold higher clustering coefficient can be observed in the B cell follicle network compared to the ER random network.

We speculate that such a complex and robust topological configuration of the underlying FDC and RC networks would presumably promote B cell-dependent processes in LN follicles as outlined in the Results section. We have amended the text (lines 165-172) to indicate the speculative nature of this statement:

“The small-world network configuration is characterised by an overabundance of highly connected nodes, common connections mediating the short mean-path lengths. This property is associated with rapid information transfer and is also observed in airline routes and social networks. In the context of the follicle, this property is likely to promote complement mediated trafficking of antigen by non-cognate B-cells from the subcapsular sinus to the follicular dendritic cell network, and also the migration of cognate B-cells as they search for antigen within the follicle, and then present it to T-cells at the interfollicular border before seeding a germinal center reaction.”

- 4. To assess the in vivo importance of cathepsin B, the authors performed a histological analysis of lymph nodes from *Ctsb* deficient mice, suggesting a defect of B-cell homing. However, the authors should provide clear in vivo demonstration that cathepsin B deficiency affects intranodal B cell migration.**
- 5. Cathepsin B deficiency may also affect T cell migration and lymph node development. How can the authors exclude this aspect?**
- 6. Do cathepsin B deficient mice have a defect in B cell adhesion in high endothelial venules, potentially explaining the reduction of LN size?**

Based on our theoretical calculations and biophysical measurements of diffusion and CXCL13 binding to extracellular matrix components, we predicted that a large fraction of CXCL13 is immobilised *in situ* and that a lack of *Ctsb*^{-/-} will lead to a lack of soluble CXCL13 gradients and aberrant spatial organisation of B cells.

To provide additional evidence to support this model and to address the reviewers concerns, we have performed additional experiments to better assess whether B-cell migration is affected, and whether aberrant T-cell migration, lymph node development, or adhesion of B-cells to high endothelial venules is contributing to our observed phenotype. These results have been incorporated in Figure 6 and S8.

Specifically, we found that all LNs are present in *Ctsb*^{-/-} mice but tend to be smaller. However, when we quantified both absolute numbers and proportions of B and T cells in lymph nodes, we did not observe statistically significant difference between WT and *Ctsb*^{-/-} mice.

To assess the impact of cathepsin B deficiency on lymph node entry and follicular homing we transferred (i.v) 1×10^7 B220⁺ CFSE⁺ cathepsin-competent CD45.2 cells into CD45.1 congenic mice from a wildtype or *Ctsb*^{-/-} background. We then assessed the spatial localisation of CFSE labelled cells within lymph nodes 24 hours later by confocal microscopy. We found no differences in the numbers of CFSE labelled cells that enter the LN, and within the LN CFSE⁺ cells were found to co-localise with other B-cells. This experiment suggests that B-cells can enter the LN and co-localise with other B-cells in the absence of soluble CXCL13 gradients.

Histological analysis of Cathepsin B deficiency (IHC with Abs against B220, CD19, PDPN, CD4, CD3, CD21/35) showed that follicles in *Ctsb*^{-/-} mice are less discrete and tend to form a thin rim that runs underneath the subcapsular sinus. In several of the mice we observed a ring-like structure as has been observed in the CXCL13 KO mouse model (Ansel et al, 2000). This was associated with reduced CD21/35 signal in the follicle.

In addition, we observed no obvious defect in T-cell localisation or in vascular morphology by histology. To corroborate these findings, we have performed RT-qPCR on whole lymph nodes looking at a panel of genes relating to glycan synthesis and the formation of PNA^d (*Meca-79*) scaffolds (*Glycam1*, *Podxl*, *Cd34*, *Madcam1*, *FuctIV*, *FuctVII*), cellular adhesion (*Icam1*, *Vcam1*, *Pecam1*) and chemokines (*Cxcl13*, *Ccl19*, *Cxcr5*, *Ccr7*). With the exception of *Podxl*, we find no statistically significant difference in deltaCT values for each gene when comparing WT and *Ctsb*^{-/-} mice. Taken together, our data suggests that cathepsin B, and by extension soluble CXCL13 gradients, are required for LN entry or follicular homing but do lead to a disruption in follicular organisation. This result is consistent with our biophysics and theoretical findings that suggest that soluble CXCL13 represents only a small fraction of total CXCL13 levels *in vivo*. We conclude that soluble CXCL13 forms only a small fraction of total CXCL13 levels and does not impact upon B-cell entry or homing to follicles but it does contribute to follicular organisation and thus function.

7. Are the results shown in Figure 2D statistically significant?

Yes, however it is important to note that traditional significance tests (t-test, anova and non-parametric equivalents) are not appropriate for comparing simulation results. The reason for this is that we could increase our sample size *n*, increasing our statistical power and allowing us to detect arbitrarily small differences between groups. For this reason, we used an effect magnitude test and point the reviewer to the details of the test included in the figure legend. To make this clearer to the reader, the A-test statistic is added directly onto the plot. More details about this test can be found in reference 56 of the manuscript.

8. The data from Figure 5J suggest some overlap between CD35+ cells and expression of cathepsin B.

Yes, this is correct. We have amended the typo that states the opposite. The amended text is now found on lines 298-99:

9. **Figure S9a mentioned in the text is missing.**
10. **Some histograms are missing from Figure N4.3.**
11. **The authors should check figure legends 3 and 4 (these two fig. legends seem inverted).**

These issues have been amended

12. **The authors mention in the discussion that “Our data thus provides a unique insight into how the primary follicle is structurally organized to promote B-cell homeostasis and activation.” However, no convincing in vivo functional data are provided to support this claim.**

We have corroborated our histology analysis in a new figure 6 and S8 with more in depth histology, flow cytometry, RT-qPCR and cellular transfer analyses as detailed above, we believe that this additional in vivo functional data is highly consistent with our biophysical and theoretical findings, helping to support our claim.

Minor points:

- **Page 13 line 306: “figure 6k” should be replaced with “figure 5k”.**
- **Line 305: “a phenotype suggestive of defective B-cell homing” is repeated two times.**
- **The significance bars need to be checked in Figure 6S.**

All aforementioned minor points have been amended according to the reviewer’s suggestions.

Reviewer #2

1. **The manuscript reproduces many findings already firmly established in the field (e.g. the production of CXCL13 by follicular dendritic and other reticular cell types and CXCL13 binding to heparin sulphate). In general the manuscript does not significantly add to the understanding of the standard “production, diffusion, immobilization, consumption”-model of chemokine function.**

The present study does not aim to re-define the classical model of chemokine function, rather it aims to validate how one chemokine can switch from soluble (mobile) and immobilized gradients through the function of an ectoenzyme to optimize B cell localization, antigen scanning and function. In addition, we

increase the classical model's quantitative rigor by enumerating “*production, diffusion, immobilization, consumption*” components where possible to understand the precise microanatomical distribution of chemokine molecules *in vivo*. This quantitative approach has enabled us to elaborate upon the CXCL13 chemokine gradient fields generated by the follicular stromal cells and in doing so, has highlighted a key role for the stromal-cell microenvironment in immobilized gradient formation. More precisely:

- (i) To date, the 3D geometry of CXCL13⁺ reticular and follicular dendritic cells had not been characterised quantitatively and analysed using a graph theory approach.
- (ii) We argue that the diffusion and immobilisation of CXCL13 is poorly understood at a quantitative level. In recent work we have shown that the diffusion constant of CXCL13 is orders of magnitude less than that predicted using the Einstein-Stokes relation which has been employed to estimate the rate of diffusion in a previously published theoretical study (Oyler Yaniv, 2017). In addition, our single molecule assay enabled us to characterise the multimodal nature of diffusion, whereby we identify distinct immobile and mobile components in the data. To the best of our knowledge, this relative contribution of mobile and immobile components has not been examined *in situ*. Furthermore, we have identified a mechanism by which cathepsin B cleaves CXCL13, mediating the switch from immobile to mobile gradients. For further details about our imaging approach we refer the reviewer to the following reference: Miller H, Cosgrove J, et al. “*High-Speed Single-Molecule Tracking of CXCL13 in the B-Follicle.*” *Frontiers in Immunology* (2018).
- (iii) The consumption component has not been addressed experimentally in this work but has been incorporated into our theoretical model. When we performed sensitivity analyses we found that B-cell migration was robust to absolute numbers of CXCR5, consistent with reports in (Coelho et al 2013) where migration was shown to be robust to a 20-fold reduction in chemokine levels associated with viral infection. There may be atypical chemokine receptors that contribute to scavenging as has been reported for CCL21, but we have not read any reports of this in the context of CXCL13.

2. Heparin-binding of CXCL13: If CXCL13 binds to heparin in follicles, how does it bind to much brighter extrafollicular heparin sulfate positive structures (possibly vessels?) in tissue sections (Fig 3A; is the numbering of Figs 3 and 4 swapped?). If the quantification of the binding of Alexa-labeled CXCL13 to tonsil tissues (Fig. 3 B) after enzymatic treatments is done from images like Fig. 3A (two lower rows) it will be biased since the extrafollicular areas are excluded. How does the Alexa-labeling affect the binding?

Yes, the legends for figures 3 and 4 were mislabelled, this has now been amended.

The capture of chemokines and decoration of luminal surfaces is a well-known phenomenon. The aim of our paper is to characterise the microarchitecture of the follicle and we make no claims about extrafollicular binding; so this experiment is out of scope. We have already provided histology and single molecule imaging data showing that binding and diffusion are perturbed when heparinase II is used to remove extracellular matrix components. The purpose of the experiment in Figure 3B is to assess follicular binding patterns. To ensure that there are no biases in the analyses, we have analysed a large number of technical replicates with several different follicles analysed.

The Alexa dye is incorporated into the C-terminus and adds an additional ~10% mass to the molecule a sufficient distance from the heparin binding domain to make it unlikely that it significantly alters binding characteristics. From our single molecule imaging assay, we identify a low mobility component in the data indicative of binding to the extracellular matrix. This is discussed further in the following reference: Miller H, Cosgrove J, et al. "*High-Speed Single-Molecule Tracking of CXCL13 in the B-Follicle.*" *Frontiers in Immunology* (2018).

In addition, the diffusion coefficient of CXCL13 increased upon treatment with heparinase II, confirming that binding to the ECM was occurring. Thus, the Alexa dye does not completely block binding to the ECM. We have added in an additional figure (Fig. 3D) showing low mobility and high mobility CXCL13 components (corresponding to matrix bound and soluble CXCL13 respectively) to highlight this.

3. The role of cathepsin-B (a well-known lysosomal enzyme) in proteolytic cleavage of CXCL13 is interesting, but very preliminary and highly speculative at this stage. The rationale for choosing cathepsin B as a putative candidate for CXCL13 degradation seems to be missing.

We have added in additional text to clarify why CXCL13 is a putative candidate for CXCL13 cleavage (lines 333-340). The amended text is as follows: Given the high affinity with which CXCL13 binds to the ECM, we hypothesized that it may require proteolytic processing to function. In this study, we focused on the cathepsin family; most cathepsins identified in humans are lysosomal enzymes involved in metabolic protein turnover but many cathepsins have also been reported to cleave chemokines^{30,31}. In particular, we have focused our attention on cathepsin B, which has been shown to regulate cytokine expression during *L. major* infection⁴¹, is upregulated in many cancers⁴², and can be produced in extracellular form in cytokine stimulated fibroblasts taken from rheumatoid arthritis patients⁴³.

The analysis of cathepsin B expression in human tissues is not very convincing (the use of polyclonal antibody in staining's would require much more stringent specificity controls; the signal in Western blotting can come from any cell type/anatomical compartment; finding of cathepsin B in supernatant of *in vitro* cultured monocytes/macrophages is not directly relevant to lymphoid tissues). In the absence of any evident alterations in signaling (Fig 5D,F), the mechanisms and *in vivo* relevance of greater potency (Fig 5E,G) of cleaved CXCL13 observed *in vitro* chemotaxis assays remain unclear. In fact, the authors do not seem to provide any evidence that such cleaved form exists *in vivo* in humans.

We agree that in isolation the IHC is insufficient to explain the source of Cathepsin B in the follicle. However, the western blotting does confirm that Cathepsin B is indeed expressed in the tissue and the *in vitro* culture system was designed specifically to validate our IHC results. Yes, we agree that in the absence of any evident alterations in signaling we cannot claim greater potency *in vivo* and do not make this claim anywhere in the manuscript. Previously we had made attempts to find the cleaved form of the molecule in human tissue using mass spectrometry, but the instrumentation was not sensitive enough to detect a rapidly cleaved very short amino acid peptide. Both our biophysical and theoretical analyses suggest that only a very small fraction of the total numbers of cleaved CXCL13 molecules are soluble making it very difficult to detect the cleaved form *in situ*.

The analysis of Cathepsin B-deficient mice is very preliminary, and from the data shown there is no way of claiming that the observed B-follicle phenotype would be linked to CXCL13 processing. For instance, the size difference of lymph nodes, and the possible alterations in other cell types, in follicular morphology and chemokine/chemokine receptor expression should be analyzed in much more detail and quantitatively.

To better address these issues we complement our histology with additional histology, flow cytometry, RT-qPCR and analyses of cell transferred as detailed above. We believe that this additional *in vivo* functional data presented in the new Figure 6 and S8 help to support our claim.

Mouse and human CXCL13 only shear about 65% similarity in the protein sequence. Does cathepsin B cleave mouse CXCL13 even *in vitro*?

By sequence alignment of the human (the mature peptide sequence is amino acids 23-109; the first 22 are signal peptides) and mouse amino acid sequences using EMBOSS needle we find that the serine at position 73 of the mature peptide is conserved across these two species. Thus, while there are some sequence differences between species, the cleavage site is conserved.

23-109	1	-----VLEVYYTSLRCRCVQESSVFIPRRFIDRI	29
		: . .: : :	
CXL13_MOUSE	1	MRLSTATLLLLLASCLSPGHGILEAHYTNLKCRCVISTVVGLENIIDRI	50
23-109	30	QILPRNGCPRKEIIVWKKNKSIVCVDPQAEWIQRMMEVLRKRS-SSTLP	78
		:. : : : . .: : : :	
CXL13_MOUSE	51	QVTPPGNGCPCKEVVIWTKMKKVICVNPRAKWLQRLLRHVQSKLSSTPQ	100
23-109	79	VPVFKRKIP	87
		. . : :..	
CXL13_MOUSE	101	APVSKRRRAA	109

Are there any defects in B-cell homing to lymph nodes or in parenchymal B-cell migration to follicles in vivo in cathepsin B knockouts?

No there are no defects in LN homing or in parenchymal B-cell migration to follicles as indicated in the amended Figure 6. To address this question, we transferred (i.v) 1×10^7 B220⁺ CFSE⁺ *Ctsb*-proficient CD45.1 B cells into CD45.2 congenic mice from a WT or *Ctsb*^{-/-} background. We then assessed the spatial localisation of CFSE labelled cells within lymph nodes 24 hours later by confocal microscopy. In this experiment, we find similar amounts of CFSE cells that enter the LN. To assess the spatial distribution of the cells within the tissue more quantitatively, we enumerated B220-CFSE co-localisation, finding no statistically significant difference between WT and KO. This experiment suggests even though follicular organisation is less robust in *Ctsb*^{-/-} mice, B-cells can enter the LN and co-localise with other B-cells consistent with biophysical and theoretical findings that the majority of CXCL13 is immobilised disrupting formation of a soluble CXCL13 gradient.

4. Much of the data are from a single patient (Figs. 3, 4).

Indeed, data in Figure 3 are representative of one patient with technical replicates, while in figure 4 (a) and (b) we show data from a single patient as an exemplar, but we do provide a quantification of data from 5 patients in figure 4c. This information can be found in the figure legends.

Reviewer #3

- 1. Page 8-9: The Pareto optimal solutions need to be explained and illustrated either in the main text or in supplementary notes.**

We had omitted a more in depth discussion in Pareto fronts as we thought this would make the paper less accessible for a wider audience but agree that more explanation would help the reader to better interpret the results. We have added a more detailed discussion about our Pareto fronts in the results section (lines 269-287).

- 2. Application of the artificial neural network-based (black-box) emulator of the multiscale (mechanistic) model rises the issue of whether its sensitivity to the parameter variations is the similar to that of the mechanistic model itself. This needs to be discussed.**

To ensure that the sensitivity of the parameter variations is similar to that of the mechanistic model itself we ran an additional Latin hypercube sampling of the parameter space to generate parameter samples that were not the same as those used to train the neural network. We then used the emulator to predict the output values for these parameter inputs and quantified parameter sensitivity for each output using the partial-rank correlation coefficient (PRCC) and compared them to those of the simulator.

This data is plotted in figure S2c and shows that the PRCC profiles of the emulator are highly similar to that of the simulator increasing confidence in the use of the emulator as a surrogate tool to reproduce key simulator behaviours.

- 3. What is the statistical framework used to assimilate the empirical data using the mechanistic model? Were the uncertainty intervals evaluated?**

For model fitting each parameter set was run 250 times (to mitigate the impact of aleatory uncertainty) and the median values of simulation responses was compared to those measured *in vivo* by means of an unpaired Mann-Whitney test.

Due to the 13-dimensional parameter space (with some parameters spanning several orders of magnitude) and the need for 250 replicate runs per parameter set to mitigate aleatory uncertainty it was not tractable to evaluate uncertainty intervals. We did however perform both local and global sensitivity analyses to quantify parametric uncertainty on each model output and our multi-objective analysis shows the trade-off between three of our model output metrics that we compare against experimental data.

- 4. Page 21: Training of artificial neural network (ANN): What is the rationale behind the selecting a given number of the hidden layers and the number of their elements. Four different structures are shown in Figure S7. Were the information-theoretic criteria used to rank the ANNs? Please, could you elaborate more on this.**

To determine suitable hyperparameters of the network, we performed ten-fold cross validation on a selection of structures with five inputs (the parameters) and three outputs (MI, MC and velocity), with one to four hidden layers. The accuracy of each fold was determined to be the root mean squared error (RMSE) between the predicted cell behaviour responses and those observed in the simulation, and the accuracy of the network structure determined to be the average of the tenfold RMSE. The network

structure with the minimum average RMSE was selected as the structure that would be used in creation of the emulator.

This is detailed briefly in the materials and methods but for further details please refer to our publication on this approach: Alden, K., J. Cosgrove, M. Coles, and J. Timmis. "Using Emulation to Engineer and Understand Simulations of Biological Systems." IEEE/ACM Transactions on Computational Biology and Bioinformatics, 2018.

5. Supplementary Note 1, page 8: the formulas for Global clustering coefficient, Average local clustering coefficient and in part, for Sigma factor are not consistent with the text and notations around them.

The formulas and notations have been amended in Supplementary Note 1 for consistency.

6. Figure 1E, length units are missing.

Shortest path lengths in Figure 1E represent a dimensionless measure and not a geographical measure of real 3D distances. They indicate shortest paths between all pairs of nodes, namely how many nodes one must transverse along the path to get from one point to the other. The legend for Figure 1E has been updated to clarify this

7. Figure 2E, Decay rate: What is the implication of the calibrated values of the parameter to be located at the edge of the curve corresponding to Pareto optimal solutions?

Due to where our calibrated value falls on the distribution any increase in the value of the decay rate will lead to an increase in scanning rates, a feature we observe in our local sensitivity analyses (note that low A-test scores are associated with higher scanning rates). This may cause us to overestimate the influence of this parameter on model behaviours and by extension on the biological system. We have added some text to discuss this in the supplementary information (bottom of p.20) to make the reader aware of this potential bias.

8. Figure 3 and Figure 4 need to be reordered to fit the legends.

9. Table N4.1: "...and parameters were removed where possible." This is not clear.

10. Figure N4.4: A,B,C are missing in the legend.

11. Figure N4.3: There are empty spaces instead of graphs.

12. Supplementary Note 4, page 19: "...the number of cells in the target location." This is not clear, i.e. the concentration or the population size are implied.

13. Supplementary Note 4, page 20: "LN(α)" is not explained. Is it a function?

The supplementary material has been amended to address points 8-13

Reviewers' comments:

Reviewer #1 (Remarks to the Author):

In this revised study, the authors claim that B-cell zone reticular cell architecture is responsible for CXCL13 gradient formation. The revised version is improved. However, there are still some remaining issues that have not been addressed as listed below.

1. Indicated lines for comment 3 are not correct.
2. In their rebuttal letter, the authors claim that they did not observe statistically significant differences in the absolute numbers of B and T cells in lymph nodes between WT and *Ctsb*^{-/-} mice. However, these results are not shown in Figure 6 or S8 and the authors need to show these data in order to exclude a role for cathepsin B in lymphocyte migration. If the absolute numbers of B and T cells are unchanged, which is the potential explanation for the reduced lymph node size in *Ctsb*^{-/-} mice?
3. The images shown in Figures 6e and 6f are not convincing. The authors need to quantify the number of migrated CSFE positive cells by flow cytometry.
4. The authors did not show the effect of cathepsin B deficiency on B cell adhesion in high endothelial venules to exclude an effect on lymphocyte adhesion, as previously requested. However, this point may become less relevant if the authors clearly demonstrate that the number of B cells inside the lymph nodes is unchanged in *Ctsb*^{-/-} mice as mentioned above at points 2 (quantification of total B cells in flow cytometry) and 3 (accumulation of CSFE-labeled B cells quantified by flow cytometry).
5. Reference 56 mentioned for A test statistic is not correct.
6. Related to my previous point 8, the authors mentioned that "The amended text is now found on lines 298-99". However, the lines need to be checked.
7. Some histograms are still missing from Figure N4.3.

Reviewer #2 (Remarks to the Author):

The authors have addressed the majority of my comments by revising the text and adding new data. However, several important issues, all pointed out in my original comment 3, still remain unanswered.

The authors conclude in the abstract that "Mice lacking cathepsin B display aberrant follicular architecture, a process dependent on soluble CXCL13". Therefore, it is essential to experimentally verify that mouse cathepsin B cleaves mouse CXCL13, and to try to exclude other potential explanations for the observed phenotype. Thus,

1. Does mouse cathepsin B cleave mouse CXCL13? This has now been addressed by analyzing sequence similarity of human and mouse CXCL13. However, the target sequences for cathepsin B remain incompletely characterized. Therefore, this question should be addressed by direct experimentation with the mouse molecules in vitro (similar to the experiment shown in Fig. 4A for the human molecules; or preferentially by showing differentially cleaved CXCL13 in lymph nodes of wt and cathepsin KO mice).
2. Comparison of wild-type and Cathepsin B knock-out mice:
 - a. The size difference of the lymph nodes. Rather than showing representative photographs (Fig. 6A), one would like to see quantifiable data (i.e. weight of the nodes). In addition, what is the brownish pigment in the inguinal nodes of KO (it is absent from wt)?

b. Follicular cell populations: More detailed analyses of the numbers of follicular and germinal center B cells, and follicular dendritic cells are needed to understand the composition of aberrant follicles.

c. CXCL13 and CXCR5 expression. When analyzed from the whole lymph nodes there seems to be a trend of decreased CXCL13 and CXCR5 mRNA expression in the KO (Fig. 6G). Were there quantifiable differences at the protein level (CXCL13 in the stromal cells by immunohistochemistry and CXCR5 on the surface of B-cells and follicular Th cells by FACS).

d. Homing experiment. Analysis of pixel co-localization is incompletely described (does it mean co-localization of CSFE+ pixels with B220+ pixels?). In any case, the standard flow cytometric analyses of homing (% of immigrated CSFE+B-cells) are needed. Possible B-cell intrinsic defects in migration to follicles should be ruled out by comparing the homing efficacy of 1:1 mixture of differentially labeled wild-type and KO B- cells to wild-type lymph nodes.

Reviewer #3 (Remarks to the Author):

In the revised manuscript the authors have productively addressed the comments on the original submission. There are minor issues (1-3) which require attention from the authors.

1. "...We have added a more detailed discussion about our Pareto fronts in the results section (lines 269-287).".

In the revised version, the text within the above range reads:

=====
=

269 together with stromal-cell network architecture, shapes complex immobilized CXCL13
270 gradients within the B cell follicle.
271
272
273 Cathepsin B-mediated formation of soluble gradient formation is required for effective
274 follicular organisation
275 Given the high affinity with which CXCL13 binds to the ECM, we hypothesized that it may
276 undergo proteolytic processing. In this study we focused on the cathepsin family; most
277 cathepsins identified in humans are lysosomal enzymes involved in metabolic protein
278 turnover but many cathepsins have also been reported to cleave chemokines^{30,31}. In particular,
279 we have focused our attention on cathepsin B (Cath-B), which has been shown to regulate
280 cytokine expression during L. major infection⁴², is upregulated in many cancers⁴³, and can be
281 produced in extracellular form in cytokine stimulated fibroblasts taken from rheumatoid
282 arthritis patients⁴⁴.
283
284 Incubation of CXCL13 with Cath-B yielded 284 two cleavage products with masses of 9.03 and
285 8.68 kDa, respectively (Figure 5a). The smaller product was stable and formed across a range
286 of enzyme substrate ratios (Figure S4a) and was detected at pH values between 4.0 and 7.2
287 with an optimal turnover rate between pH 5.0 and 6.5 (Figure S4b). Consistent with this data,...

=====
=

Something is wrong with the line numbers.

2. The references in SN4 need to be checked. They do not match the meaning of citations.
(e.g.,

"...This mathematical construct is capable of isotropic diffusion [1]..."

[1] Junt, T., Scandella, E. & Ludewig, B. Form follows function: lymphoid tissue microarchitecture in antimicrobial immune defence. *Nat. Rev. Immunol.* 8, 764–775 (2008).;

"...with diffusion modeled using a discretized partial differential equation (Module 2) [51]", whereas

[51] Plank, M., Wadhams, G. H. & Leake, M. 852 C. Millisecond timescale slimfield imaging and automated quantification of single fluorescent protein molecules for use in probing complex biological processes. *Integr. Biol. Quant. Biosci. Nano Macro* 1, 602–612 (2009).);
etc.)

3. Figure N4.3. has missing parts.

Reviewer #1 (Remarks to the Author): In this revised study, the authors claim that B-cell zone reticular cell architecture is responsible for CXCL13 gradient formation. The revised version is improved. However, there are still some remaining issues that have not been addressed as listed below.

1. Indicated lines for comment 3 are not correct. **The indicated lines for comment 3 can be found on lines 174-181 in the latest version of the manuscript**

2. In their rebuttal letter, the authors claim that they did not observe statistically significant differences in the absolute numbers of B and T cells in lymph nodes between WT and *Ctsb*^{-/-} mice. However, these results are not shown in Figure 6 or S8 and the authors need to show these data in order to exclude a role for cathepsin B in lymphocyte migration. If the absolute numbers of B and T cells are unchanged, which is the potential explanation for the reduced lymph node size in *Ctsb*^{-/-} mice? **We have now repeated the transfer experiments with new mice and analysed absolute numbers of B cells entering into LNs (Figure 6g) of WT and Cathepsin B KO mice and relative efficiency in entry into LNs and spleen (Figure 6f, Figure S10). We find no statistical difference in the homing rates of WT vs *Ctsb*^{-/-} mice to LNs. We posit that due to aberrant follicular homing within the lymph node B-cell associated stromal cell networks are on average smaller and less mature than WT counterparts (see figure 6c) – and that this is the reason *Ctsb*^{-/-} LNs are smaller in size.**

3. The images shown in Figures 6e and 6f are not convincing. The authors need to quantify the number of migrated CFSE positive cells by flow cytometry. **This has been performed and is included in the main manuscript as figure 6f-g.**

4. The authors did not show the effect of cathepsin B deficiency on B cell adhesion in high endothelial venules to exclude an effect on lymphocyte adhesion, as previously requested. However, this point may become less relevant if the authors clearly demonstrate that the number of B cells inside the lymph nodes is unchanged in *Ctsb*^{-/-} mice as mentioned above at points 2 (quantification of total B cells in flow cytometry) and 3 (accumulation of CFSE-labeled B cells quantified by flow cytometry). **This issue has been addressed by our quantification of CFSE-labelled B cell homing into LNs by flow cytometry (Figure 6f-g).**

5. Reference 56 mentioned for A test statistic is not correct. **This reference has been updated. Additionally, the same reference in the supplementary material has now been updated and an additional bibliography has been added to the supplementary material to avoid any inconsistencies.**

6. Related to my previous point 8, the authors mentioned that “The amended text is now found on lines 298-99”. **However, the lines need to be checked. In the current version of the manuscript this can be found on lines 320-321.**

7. Some histograms are still missing from Figure N4.3. **This issue has been addressed.**

Reviewer #2 (Remarks to the Author): The authors have addressed the majority of my comments by revising the text and adding new data. However, several important issues, all pointed out in my original comment 3, still remain unanswered. **We thank the reviewer for their input and through the provision of two new sets of data are able to address their concerns. Specifically, we have verified experimentally that cathepsin B cleaves mouse CXCL13 and through B-cell transfer experiments have ruled out other hypotheses that could be used to explain the phenotype of *Ctsb*^{-/-} mice.**

The authors conclude in the abstract that “Mice lacking cathepsin B display aberrant follicular architecture, a process dependent on soluble CXCL13”. Therefore, it is essential to experimentally verify that mouse cathepsin B cleaves mouse CXCL13, and to try to exclude other potential explanations for the observed phenotype. Thus:

1. Does mouse cathepsin B cleave mouse CXCL13? This has now been addressed by analyzing sequence similarity of human and mouse CXCL13. However, the target sequences for cathepsin B remain incompletely characterized. Therefore, this question should be addressed by direct experimentation with the mouse molecules *in vitro* (similar to the experiment shown in Fig. 4A for the human molecules; or preferentially by showing differentially cleaved CXCL13 in lymph nodes of wt and cathepsin KO mice). **We have verified this experimentally and can confirm that cathepsin B does cleave mouse CXCL13 – the updated data are provided in figure S4a.**

2. Comparison of wild-type and Cathepsin B knock-out mice:

a. The size difference of the lymph nodes. Rather than showing representative photographs (Fig. 6A), one would like to see quantifiable data (i.e. weight of the nodes). In addition, what is the brownish pigment in the inguinal nodes of KO (it is absent from wt)? **We consistently observed smaller LNs controlling for mouse sex, age and colony location (mice from colony of *Ctsb*^{-/-} in Paris and Oxford). Overall we found the most consistent measure was to use absolute numbers of B cells as a quantification of overall size which is shown in Figure S11. We did not consistently find the pigment although it is known in mice from the C57BL6 background that melanosis can occur in secondary lymphoid tissues being particularly common in the spleen of younger mice, this is a potential explanation for the colouration difference but as it was not consistently observed (as is found with melanosis) thus we do not think it is important in the phenotype of the mice. The role of cathepsin B in melanin production is unknown and not thought to be of immunological significance. Alternative explanations could include haemosiderosis (iron induced hemosiderin accumulation) or lipofuscinosis (lipidpigment accumulation) are unlikely and would be unrelated to the immunological phenotype observed.**

b. Follicular cell populations: More detailed analyses of the numbers of follicular and germinal center B cells, and follicular dendritic cells are needed to understand the composition of aberrant follicles. **We have now quantified absolute numbers of WT and KO B cells (Figure 6f). As the purpose of this study is to understand the spatial distribution of CXCL13 in the *primary* follicle we argue that a detailed analysis of germinal center B cells is out of scope. Given the substantial remodelling of the B-cell microenvironment that occurs in secondary follicles it is not appropriate to extrapolate our findings to the germinal center reaction.**

c. CXCL13 and CXCR5 expression. When analyzed from the whole lymph nodes there seems to be a trend of decreased CXCL13 and CXCR5 mRNA expression in the KO (Fig. 6G). Were there quantifiable differences at the protein level (CXCL13 in the stromal cells by immunohistochemistry and CXCR5 on the surface of B-cells and follicular Th cells by FACS). **We could only get robust immunohistochemistry results for human CXCL13 and found that the commercially available mouse Abs were binding non-specifically, or else the signal obtained was barely different from background noise in mouse tissues, although in the past we have had some success, more recent batches of antibodies in mouse tissue sections often did not work. In previous work performed in the lab, we have also not been able to get consistent results with CXCR5 Abs for mice, despite efforts to optimise our working protocols. Consequently, we have decided not to include any data at the protein level in mouse**

tissues (human data is still included), as the results were not consistent enough for us to draw any reliable conclusions

d. Homing experiment. Analysis of pixel co-localization is incompletely described (does it mean co-localization of CSFE+ pixels with B220+ pixels?). In any case, the standard flow cytometric analyses of homing (% of immigrated CSFE+B-cells) are needed. Possible B-cell intrinsic defects in migration to follicles should be ruled out by comparing the homing efficacy of 1:1 mixture of differentially labeled wild-type and KO B- cells to wild-type lymph nodes. We have performed this experiment exactly as suggested by the reviewers labelling B cells from Cathepsin B KO mice with CSFE and WT B cells with CMTMR and transferred into WT or Cathepsin B KO mice and analysed by flow cytometry (6f-g). We find no significant difference in LN homing in either the WT or *Ctsb*^{-/-} setting and conclude that there are no obvious defects in homing efficacy.

Reviewer #3 (Remarks to the Author): In the revised manuscript the authors have productively addressed the comments on the original submission. There are minor issues (1-3) which require attention from the authors. We thank the reviewer for their very helpful corrections. We have made corrections to line numbers, references and figure N4.3.

1. "...We have added a more detailed discussion about our Pareto fronts in the results section (lines 269-287)." The updated description of the Pareto front of solutions is provided in 224-228.

2. The references in SN4 need to be checked. They do not match the meaning of citations. (e.g., "...This mathematical construct is capable of isotropic diffusion [1]..." [1] Junt, T., Scandella, E. & Ludewig, B. Form follows function: lymphoid tissue microarchitecture in antimicrobial immune defence. *Nat. Rev. Immunol.* 8, 764–775 (2008); "...with diffusion modeled using a discretized partial differential equation (Module 2) [51]", whereas [51] Plank, M., Wadhams, G. H. & Leake, M. 852 C. Millisecond timescale slimfield imaging and automated quantification of single fluorescent protein molecules for use in probing complex biological processes. *Integr. Biol. Quant. Biosci. Nano Macro* 1, 602–612 (2009).; etc.). We have added a distinct bibliography for the supplementary material to avoid any inconsistencies in our references.

3. Figure N4.3. has missing parts. This was a formatting error and has been amended.

Reviewers' comments:

Reviewer #1 (Remarks to the Author):

The authors have responded to the majority of questions. However, there are some points which need to be addressed and errors to be corrected.

a. Each point in Figures 6f and g represents a single lymph node taken from a distinct mouse with 4-5 mice per group. However, if the in vivo data are from one single experiment with one lymph node/mouse, did the authors repeated these experiments? The authors should clearly state how many experiments were performed for these figures. Also, it is unclear how many experiments were performed for the results shown in Supplementary Figures 10 and 11. This is particularly important, as the results shown in Supplementary Figure 11 indicate a reduction of B cell accumulation in LNs from *Ctsb*^{-/-} mice; these data did not reach statistical significance probably due to a reduced number of lymph nodes/animals/experimental condition and/or insufficient number of experiments.

b. Figure legends should be provided for Supplementary Figures 10 and 11 with the description of the experiments.

c. Figure legend 6 needs to be corrected as Fig. 6G now shows homing data and not PCR results. Figure H is missing from the Figure legend.

d. The number of cells seem similar between WT and *Ctsb*^{-/-} mice in Fig. 6E. However, the authors should discuss in the paper the results presented in Fig. 6 E, which shows that CSFE positive cells clearly overlap with B220 positive areas in WT animals, whereas these cells are more disperse and migrate also in B220 negative zones in *Ctsb*^{-/-} mice.

e. Related to point 6 (former point 8), lines 320-321 are not correctly indicated by the authors and do not address my point.

Reviewer #2 (Remarks to the Author):

The revisions have improved the manuscript.

However, the statement (in the Abstract) that cathepsin B-mediated production of soluble CXCL13 would explain the aberrant follicular architecture still remains speculative, since the lymph node size, architecture and homing are apparently quite variable in these experiments. The statement could be tuned-down (as already has been done in the Results section).

Cell transfer experiments: Only highly processed data (ko:wt ratios based on some normalization to wt CSFE:wt CMTMR ratio) are shown. It would be useful to comment (at least in the Methods) about the frequency of green and red cells in the input population and the absolute numbers of labeled cells recovered from the lymph node.

Fig 6 legend panels. Panel g: It should be clarified that this has been done with FACS (if I got it right), and the current legend of g probably refers to h)

Suppl Fig. 11 would benefit from a legend. Are these data from untouched mice or from the cell-transfer experiments?

Response to Specific Reviewers comments:

Reviewer #1 (Remarks to the Author):

The authors have responded to the majority of questions. However, there are some points which need to be addressed and errors to be corrected.

*a. Each point in Figures 6f and g represents a single lymph node taken from a distinct mouse with 4-5 mice per group. However, if the in vivo data are from one single experiment with one lymph node/mouse, did the authors repeated these experiments? The authors should clearly state how many experiments were performed for these figures. Also, it is unclear how many experiments were performed for the results shown in Supplementary Figures 10 and 11. This is particularly important, as the results shown in Supplementary Figure 11 indicate a reduction of B cell accumulation in LNs from *Ctsb*^{-/-} mice; these data did not reach statistical significance probably due to a reduced number of lymph nodes/animals/experimental condition and/or insufficient number of experiments.*

Experiments for figures 6f,g and supplementary figures 10 and 11 were performed once. However, in a completely separate experiment where we quantify B-cell numbers and proportions in lymph nodes (Figure 6b shows the proportions, data generated for our 2nd submission) we also did not observe a statistically significant change in B-cell numbers (please refer to figure 1 attached below). Due to differences in experimental design between this experiment, and the experiment from Figure S11 it is not possible to pool the data. This data has now been incorporated into Figure S11 for clarity.

We do agree that in figure S11 there is a definite trend indicating a reduction of B-cell accumulation, but this is largely driven by the presence of an outlier data-point. We investigated this further and found an effect size of 0.65 million cells difference between groups, well within the variability expected from a WT mouse. Power calculations on the same dataset, show that to see a statistically significant difference (alpha = 0.05, power = 0.95) at this effect size we would need 65 mice per group.

Given that this was a negative result, was repeated under a slightly different context, the amount of mice required to achieve sufficient statistical power, and a lack of clarity about how biologically meaningful such a small change would be we do not think that this it is appropriate to repeat this experiment and don't believe it will change any of the outcomes from the manuscript.

Figure 1: A separate experiment where we quantify B-cell numbers in WT vs *Ctsb*^{-/-} mice by flow cytometry. This data is presented as % in Figure 6b in the current version of the manuscript.

b. Figure legends should be provided for Supplementary Figures 10 and 11 with the description of the experiments.

We have updated the figure legends to provide a more accurate description of the experiments

c. Figure legend 6 needs to be corrected as Fig. 6G now shows homing data and not PCR results. Figure H is missing from the Figure legend.

The figure legend has now been updated

d. The number of cells seem similar between WT and *Ctsb*^{-/-} mice in Fig. 6E. However, the authors should discuss in the paper the results presented in Fig. 6 E, which shows that CFSE positive cells clearly overlap with B220 positive areas in WT animals, whereas these cells are more disperse and migrate also in B220 negative zones in *Ctsb*^{-/-} mice.

We have included the following text in lines 350-352:

“In addition, confocal microscopy of LN sections shows that while CFSE+ cells clearly overlap with B220+ areas of WT animals, CFSE+ cells are much more disperse and are found more frequently in B220 negative zones in *Ctsb*^{-/-} mice. “

e. Related to point 6 (former point 8), lines 320-321 are not correctly indicated by the authors and do not address my point.

Your point is: **“The data from Figure 5J suggest some overlap between CD35+ cells and expression of cathepsin B.”**

We agree with this and have included an explicit statement on this which is on lines 321-323: “To determine if Cath-B was expressed in the follicle we performed IHC of tonsil tissue, with signal observed throughout the follicle, with highest expression co-localised with CD68+ cells and some co-expression on CD35+ stromal cells (Figure 5h,5i).”

Reviewer #2 (Remarks to the Author):

The revisions have improved the manuscript.

However, the statement (in the Abstract) that cathepsin B-mediated production of soluble CXCL13 would explain the aberrant follicular architecture still remains speculative, since the lymph node size, architecture and homing are apparently quite variable in these experiments. The statement could be tuned-down (as already has been done in the Results section).

The abstract has been reworded to better address this issue, and the specific statement in question now reads: “Mice lacking cathepsin B display aberrant follicular architecture, a phenotype associated with effective B cell homing to but not *within* lymph nodes.”

Cell transfer experiments: Only highly processed data (ko:wt ratios based on some normalization to wt CFSE:wt CMTMR ratio) are shown. It would be useful to comment (at least in the Methods) about the frequency of green and red cells in the input population and the absolute numbers of labeled cells recovered from the lymph node.

When performing cell transfer experiments, the authors had found that the efficiency of cell entry was affected by a more general effect that CFSE labelled cells irrespective of haplotype were more

efficient at LN entry, we suspect these results (despite 1:1 mixture) from a slight effect of cell viability post transfer of CMTMR labelled cells. The reasons for this are not known but have been observed previously in the group. Thus, interpretation of the results would otherwise been skewed by the fact that knock-out cells were CFSE labelled and WT were CMTMR labelled and results would have shown that knock-out cells were more efficient at cell entry. As we knew this resulted from the effect of the relative effect of CMTMR on cell viability we used control mice where the relative efficiency of CFSE and CMTMR labelling was measured to process the data, so this artificial effect was removed. Once this processing was undertaken no net difference was found in cellular migration in knock-out and wild type B cells. These results are consistent with the phenotype resulting from effects on intra-node B cell migration guided by CXCL13 rather than B cell entry into lymph nodes. This is consistent with the phenotypes of CXCR5 and CXCL13 deficient mice.

We have modified the materials and methods section to explicitly state how this was performed.

Fig 6 legend panels. Panel g: It should be clarified in the figure legend that this has been done with FACS (if I got it right), and the current legend of g probably refers to h)

This issue has been addressed with updated figure legend and methods section.

Suppl Fig. 11 would benefit from a legend. Are these data from untouched mice or from the cell-transfer experiments?

A more precise figure legend has been added, these data are from the cell transfer experiments. We have also included cell counts from a different experiment where untouched mice (TW vs KO) are compared. We have added an explicit statement on this in the figure legend for clarity.

REVIEWERS' COMMENTS:

Reviewer #1 (Remarks to the Author):

In this revised version, the authors addressed my concerns in a satisfactory manner. The line numbers mentioned in the rebuttal letter do not correspond in the merged file containing the manuscript text, but this is probably due to the conversion to the pdf.